# The use of QBO, ENSO and NAO perturbations in the evaluation of GOME-2/MetopA total ozone measurements

Kostas Eleftheratos[1,2], Christos S. Zerefos[2,3,4,5], Dimitris S. Balis[6], Maria-Elissavet Koukouli[6], John Kapsomenakis[3], Diego G. Loyola[7], Pieter Valks[7], Melanie Coldewey-Egbers[7], Christophe Lerot[8], Stacey M. Frith[9], Amund S. Haslerud[10], Ivar S. A. Isaksen[10,11], Seppo Hassinen[12]

[1]Laboratory of Climatology and Atmospheric Environment, Faculty of Geology and Geoenvironment, National and Kapodistrian University of Athens, Greece
[2]Biomedical Research Foundation of the Academy of Athens, Athens, Greece
[3]Research Centre for Atmospheric Physics and Climatology, Academy of Athens, Athens, Greece
[4]Mariolopoulos-Kanaginis Foundation for the Environmental Sciences, Athens, Greece
[5]Navarino Environmental Observatory (N.E.O.), Messinia, Greece
[6]Laboratory of Atmospheric Physics, Department of Physics, Aristotle University of Thessaloniki, Greece
[7]Institut für Methodik der Fernerkundung (IMF), Deutsches Zentrum für Luft- und Raumfahrt (DLR), Oberpfaffenhofen, Germany
[8]Royal Belgian Institute for Space Aeronomy (BIRA), Brussels, Belgium
[9]Science Systems and Applications, Inc., Lanham, MD, USA
[10]Cicero Center for International Climate Research, Oslo, Norway
[11]Department of Geosciences, University of Oslo, Oslo, Norway
[12]Finnish Meteorological Institute, Helsinki, Finland

*Correspondence to*: Kostas Eleftheratos (kelef@geol.uoa.gr)

**Abstract.** In this work we present evidence that quasi cyclical perturbations in total ozone (Quasi Biennial Oscillation (QBO), El Nino Southern Oscillation (ENSO) and North Atlantic Oscillation (NAO)) can be used as independent proxies in evaluating Global Ozone Monitoring Experiment-2 aboard MetopA (GOME-2A) satellite total ozone data, using ground-based measurements, other satellite data and chemical transport model calculations. The analysis is performed in the frame of the validation strategy on longer time scales within the European Organisation for the Exploitation of Meteorological Satellites (EUMETSAT), Satellite Application Facility on Atmospheric Composition Monitoring (AC SAF) project, and covers the period 2007-2016. Comparison of GOME-2A total ozone with ground observations shows mean differences of about –0.7 ± 1.4% in the tropics (0-30 deg.), about +0.1 ± 2.1% in mid-latitudes (30-60 deg.), and about +2.5 ± 3.2% and 0.0 ± 4.3% over the northern and southern high latitudes (60-80 deg.), respectively. In general, we find that GOME-2A total ozone data depict the QBO/ENSO/NAO natural fluctuations in concurrence with co-located Solar Backscatter Ultraviolet Radiometer (SBUV), GOME-type Total Ozone Essential Climate Variable (GTO-ECV; composed of total ozone observations from GOME (Global Ozone Monitoring Experiment), SCIAMACHY (SCanning Imaging Absorption SpectroMeter for Atmospheric CHartographY), GOME-2A, and OMI (Ozone Monitoring Instrument) combined into one homogeneous time series) and ground-based (GB) observations. Total ozone from GOME-2A is well correlated with the QBO (highest correlation in the tropics of +0.8) in agreement with SBUV, GTO-ECV and GB data which also give the highest correlation in the tropics. The differences between deseazonalised GOME-2A and GB total ozone in the tropics are within ±1%. These differences were tested further as to their correlations with the QBO. The differences had practically no QBO signal, providing an independent test of the stability of the long-term variability

of the satellite data. Correlations between GOME-2A total ozone and the Southern Oscillation Index (SOI) were
studied over the tropical Pacific Ocean after removing seasonal, QBO and solar cycle related variability.
Correlations between ozone and SOI are on the order of +0.5, consistent with SBUV and GB observations.
Differences between GOME-2A and GB measurements at the station of Samoa (American Samoa; 14.25$^{\circ}$ S, 170.6$^{\circ}$
W) are within ±1.9%. We also studied the impact of NAO on total ozone in the northern mid-latitudes in winter. We
find very good agreement between GOME-2A and GB observations over Canada and Europe as to their NAO-
related variability, with mean differences reaching the ±1% levels. The agreement and small differences which were
found between the independently produced total ozone data sets as to the influence of QBO, ENSO and NAO show
the importance of these climatological proxies as additional tool for monitoring the long-term stability of satellite-
ground truth biases.
**1 Introduction**
Ozone is an important gas of the Earth's atmosphere. In the stratosphere, ozone is considered *good ozone* because it
absorbs ultraviolet-B radiation from the Sun, thus protecting the biosphere from a large part of the Sun's harmful
radiation (e.g. Eleftheratos et al., 2012; Hegglin et al., 2015). In the lower atmosphere and near the surface, natural
ozone has an equally important beneficial role because it initiates the chemical removal of air pollutants from the
atmosphere such as carbon monoxide, nitrogen oxides and methane. Above natural levels however, ozone is
considered *bad ozone* because it can harm humans, plants and animals. In addition, ozone is a greenhouse gas,
warming the Earth's surface. In both the stratosphere and the troposphere, ozone absorbs infrared radiation emitted
from Earth's surface, trapping heat in the atmosphere. As a result, increases or decreases in stratospheric or
tropospheric ozone induce a climate forcing (Hegglin et al., 2015).
Ozone in the atmosphere can be measured by ground-based instruments, balloons, aircraft and satellites and can be
calculated by chemical transport model (CTM) simulations. Measurements by satellites from space provide ozone
profiles and column amounts over nearly the entire globe on a daily basis (e.g. WMO, 2014). The three Global
Ozone Monitoring Experiment-2 (GOME-2) instruments carried on Metop platforms A, B and C serve this purpose.
The first was launched on 19 October 2006, the second on 19 September 2012 and the last on 7 November 2018.
The three GOME-2 instruments will provide unique long-term data sets of more than 15 years (2007-2024) related
to atmospheric composition and surface ultraviolet radiation using consistent retrieval techniques (Hassinen et al.,
2016). The GOME-2 off-line data is set to make a significant contribution towards climate and atmospheric research
while providing near real-time data for use in weather forecasting and air quality forecasting applications (Hassinen
et al., 2016).
Validation of satellite ozone measurements is performed with ground-based (GB) measurements as well as other
satellite instruments (Hassinen et al., 2016). Validation of GOME-2A total ozone for the period 2007-2011 was
performed by Loyola et al. (2011) and Koukouli et al. (2012). It was found that GOME-2 total ozone data agree at
the ±1% level with GB measurements and other satellite data sets (Hassinen et al., 2016). The consistency between
GOME-2A and GOME-2B total ozone columns, including a validation with GB measurements, was presented by
Hao et al. (2014). An updated time series of the differences between GOME-2A and GOME-2B with GB
observations can be found in Hassinen et al. (2016). The long-term stability of the two satellite instruments was also
noted in that study. Both satellites are consistent over the Northern Hemisphere with negligible latitudinal
dependence, while over the Southern Hemisphere there is a systematic difference of 1% between the two satellite
instruments (Hassinen et al., 2016).
Chiou et al. (2014) compared zonal mean total column ozone inferred from three independent multi-year data
records, namely, SBUV (v8.6) total ozone (McPeters et al., 2013), GOME-type Total Ozone Essential Climate
Variable (GTO-ECV) (Coldewey-Egbers et al., 2015; Garane et al., 2018), and GB total ozone for the period 1996-
2011. Their analyses were conducted for the latitudinal zones of 0-30$^o$ S, 0-30$^o$ N, 50-30$^o$ S, and 30-60$^o$ N. It was
found that, on average, the differences in monthly zonal mean total ozone vary between -0.3 and 0.8% and are well
within 1%. In that study it was concluded that despite the differences in the satellite sensors and retrievals methods,
the SBUV v8.6 and GTO-ECV data records show very good agreement both in the monthly zonal mean total ozone
and the monthly zonal mean anomalies between 60°S and 60°N. The GB zonal means showed larger scatter in the
monthly mean data compared to satellite-based records, but the scatter was significantly reduced when seasonal
zonal averages were analysed. The differences between SBUV and GB total ozone data presented in Chiou et al.
(2014) are well in agreement with Labow et al. (2013), who systematically compared SBUV (v8.6) total ozone data
with that measured by Brewer and Dobson instruments at various stations as a function of time, satellite solar zenith
angle, and latitude. The comparisons showed good agreement (within ±1%) over the past 40 years with very small
bias approaching zero over the last decade. Comparisons with ozone sonde data showed good agreement in the
integrated column up to 25 hPa with differences not exceeding 5% (Labow et al., 2013).
The observed small biases (at the percentage level) between satellite and GB observations of total ozone, as have
been documented in the above studies, ensure the provision of accurate satellite ozone measurements. The high
accuracy and stability of the satellite instruments is essential for monitoring the expected recovery of the ozone layer
resulting from measures adopted by the 1987 Montreal protocol and its amendments (e.g., Zerefos et al., 2009;
Loyola et al., 2011; Solomon et al., 2016; de Laat et al., 2017; Kuttippurath and Nair, 2017; Pazmiño et al., 2018;
Stone et al., 2018; Strahan and Douglass, 2018). It is known that total ozone varies strongly with latitude and
longitude as a result of chemical and transport processes in the atmosphere. Total ozone also varies with season.
Seasonal variations are larger over middle and high latitudes and smaller in the tropics (e.g. WMO, 2014). On longer
time scales total ozone variability is related to large scale natural oscillations such as the Quasi-Biennial Oscillation
(QBO) (e.g. Zerefos et al., 1983; Baldwin et al., 2001), the El Nino Southern Oscillation (ENSO) (e.g. Zerefos et al.,
1992; Oman et al., 2013; Coldewey-Egbers et al., 2014), the North Atlantic Oscillation (NAO) (e.g. Ossó et al.,
2011; Chehade et al., 2014) and the 11-year solar cycle (e.g. Zerefos et al., 2001; Tourpali et al., 2007; Brönniman et
al., 2013). Moreover, volcanic eruptions may also alter the thickness of the ozone layer (Zerefos et al., 1994;
Frossard et al., 2013; Rieder et al., 2013; WMO, 2014). These natural perturbations affect the background
atmosphere and consequently the distribution of the ozone layer. In this context, the study of the effect of known
natural fluctuations in total ozone could serve as additional tool for evaluating the long-term variability of satellite
total ozone data records.
The objective of the present work is to examine the ability of the GOME-2A total ozone data to capture the
variability related to dynamical proxies of global and regional importance such as the QBO, ENSO and NAO, in
comparison to GB measurements, other satellite data and model calculations. The variability of total ozone from
GOME-2A is compared with the variability of total ozone from the other examined data sets during these naturally-
occurring fluctuations in order to evaluate the ability of GOME-2A to depict natural perturbations. The analysis is
performed in the frame of the validation strategy of GOME-2A data on longer time scales within the project of
EUMETSAT, AC SAF. The evaluation of GOME-2A data performed here includes the study of monthly means of
total ozone, the annual cycle of total ozone, the amplitude of the annual cycle [i.e., (max-min)/2], the relation with
the QBO (correlation with zonal wind at the equator at 30 hPa), the relation with ENSO (correlation with SOI) and
the relation with the NAO (correlation with the NAO index in winter (DJF mean)).
The annual cycle describes regular oscillations in total ozone that occur from month to month within a year. In
general, month-to-month variations of total ozone are larger in middle and high latitudes than in the tropics. The
QBO dominates the variability of the equatorial stratosphere (~16-50 km) and is easily seen as downward
propagating easterly and westerly wind regimes, with a variable period averaging approximately 28 months.
Circulation changes induced by the QBO affect temperature and chemistry (Baldwin et al., 2001). ENSO and NAO
are naturally-occurring patterns or modes of atmospheric and oceanic variability, which orchestrate large variations
in climate over large regions with profound impacts on ecosystems (Hurrell and Deser, 2009). We present the level
of agreement between satellite-derived GOME-2A and GB total ozone in depicting natural oscillations like QBO,
ENSO and NAO, highlighting the importance of these climatological proxies to be used as additional tools for
monitoring the long-term stability of satellite-ground truth biases.
**2 Data sources**
The analysis uses GOME-2 satellite total ozone columns for the period 2007-2016. This data forms part of the
operational EUMETSAT AC SAF GOME-2/MetopA GDP4.8 data product provided by the German Aerospace
Center (DLR). The GOME-2 total ozone data have been monthly averaged on a $1^o \times 1^o$ latitude longitude grid. The
overview of the GOME-2A satellite instrument and of the GOME-2 atmospheric data provided by AC SAF can be
found in Hassinen et al. (2016).
To examine the natural variability of ozone on longer time scales, we have additionally analysed the GOME/ERS-2,
SCIAMACHY/Envisat, GOME-2A, and OMI/Aura merged prototype level 3 harmonized data record (GTO-ECV,
$1^o \times 1^o$) for the period 1995-2016 (Coldewey-Egbers et al., 2015; Garane et al., 2018). This GTO-ECV ozone data
product was generated and provided by DLR as part of the European Space Agency Ozone Climate Change
Initiative (ESA O3 CCI) project. The ESA O3 CCI merged level-3 record, which is based on
GOME/SCIAMACHY/GOME-2A/OMI level-2 data, was obtained using the GODFIT v3.0 retrieval algorithm.
More on ESA O3 CCI datasets can be found in the studies by Van Roozendael et al. (2012), Lerot et al. (2014),
Koukouli et al. (2015) and Garane et al. (2018).
Both datasets are compared with a combined TOMS/OMI/OMPS satellite total ozone data set constructed using data
from the Total Ozone Mapping Spectrometer (TOMS) on Nimbus 7 (1979-1993), TOMS on Meteor 3 (1991-1994),
TOMS on Earth Probe (1996-2005), the Ozone Monitoring Instrument (OMI) onboard the NASA Earth Observing
System (EOS) Aura satellite (2005-present) and data from the next generation Ozone Mapping Profiler Suite
(OMPS) nadir profiler instrument, launched in October 2011 on the Suomi National Polar-orbiting Partnership
(NPP) satellite (McPeters et al., 2015). The total ozone data are available at $1^o$ x $1.25^o$ (TOMS) or $1^o$ x $1^o$
(OMI/OMPS) resolution from https://acd-ext.gsfc.nasa.gov/anonftp/toms/ (last access: 15 June 2018). From these
data we constructed monthly mean total ozone data on a $5^o$ x $5^o$ grid. To account for known biases between the
instruments (e.g., Labow et al., 2013) we use the Solar Backscatter Ultraviolet (SBUV) version 8.6 Merged Ozone
Data Set (MOD) monthly zonal mean total ozone (https://acd-ext.gsfc.nasa.gov/Data_services/merged/index.html,
also see next paragraph; last access: 15 June 2018) as a reference. We adjust each instrument such that the zonal
mean in each $5^o$ band averaged over the instrument lifetime matches the corresponding SBUV MOD zonal mean
average. Thus the inherent longitudinal variability is retained from the TOMS/OMI/OMPS measurements but any
latitude-dependent bias between the instruments is removed. With the exception of Meteor 3 TOMS in the northern
hemisphere, all offsets were within 2% at low and mid-latitudes. Such a data set should not be used for long-term
trends but is sufficient for analyzing periodic variability such as QBO, ENSO and NAO. We used data for the period
1995-2016. We note here that another long-term data set which has been analysed for QBO, ENSO, NAO and other
perturbations comes from the Multi-Sensor Reanalysis (Knibbe et el., 2014), but is not examined here.
In addition, we compare with satellite SBUV station overpass data from 1995 to 2016. The satellite data are based
on measurements from three SBUV-type instruments from April 1970 to the present (continuous data coverage from
November 1978). Even though the time series includes different versions of the SBUV instrument, the basic
measurement technique remains the same over the advancement of the instrument from the Backscatter Ultraviolet
(BUV) to SBUV/2 (Bhartia et al., 2013). Satellite overpass data over various ground-based stations are provided per
day from https://acd-ext.gsfc.nasa.gov/anonftp/toms/sbuv/MERGED/ (last access: 15 June 2018). These overpass
data are analogous to the SBUV MOD monthly zonal mean data previously mentioned. Both are constructed by first
filtering lesser quality measurements and then averaging data from individual satellites when more than one
instrument is operating. Monthly averages have been calculated by averaging the daily merged ozone overpass data
for stations listed in Supplement Table S1. Details about the data are provided by McPeters et al. (2013) and Frith et
al. (2014).
We also compare with GB observations of total ozone from a number of stations contributing to the World Ozone
and Ultraviolet Radiation Data Centre (WOUDC). The WOUDC data centre is one of six World Data Centres which
are part of the Global Atmosphere Watch programme of the World Meteorological Organization (WMO). The
WOUDC data centre is operated by the Meteorological Service of Canada, a branch of Environment Canada. In
total, we analysed total ozone daily summaries from 193 ground-based stations operating either Brewer, Dobson,
filter, SAOZ or microtops instruments. The GB total ozone measurements are available from the website
https://woudc.org/archive/Summaries/TotalOzone/Daily_Summary/ (last access: 15 June 2018). The various stations
used in this study are listed in Table S1.
We have also analysed simulations of total ozone from the global 3-D chemical transport model (CTM) Oslo CTM3
(Søvde et al., 2012). The Oslo CTM3 has traditionally been driven by 3-hourly meteorological forecast data from
the European Centre for Medium-Range Weather Forecasts (ECMWF) Integrated Forecast System (IFS) model,
whereas in this study we apply the OpenIFS model (https://software.ecmwf.int/wiki/display/OIFS/) (last access: 15
June 2018), cycle 38r1, which is an improvement from Søvde et al. (2012). Details on the model are given in Søvde
et al. (2012). The Oslo CTM3 comprises both detailed tropospheric and stratospheric chemistry. Photochemistry is
calculated using fast-JX version 6.7c (Prather, 2012), and chemical kinetics from JPL 2010 (Sander et al., 2011).
Total ozone columns compare well with measurements and other model studies (Søvde et al., 2012 and references
therein). The horizontal resolution of the model is $2.25^{o}$ x $2.25^{o}$. We used the global monthly mean total ozone
columns for the period 1995-2016.
To examine the QBO component on total ozone we made use of the monthly mean zonal winds at Singapore at 30
hPa. The zonal wind data at 30 hPa were provided by the Freie Universität Berlin (FU-Berlin) at http://www.geo.fu-
berlin.de/met/ag/strat/produkte/qbo/qbo.dat (last access: 15 June 2018) (Naujokat, 1986). The impact of ENSO in
the tropics was investigated by using the Southern Oscillation Index (SOI) from the Bureau of Meteorology of the
Australian Government (http://www.bom.gov.au/climate/current/soi2.shtml) (last access: 15 June 2018). The
correlation between total ozone and the NAO index was mainly computed for the winter-mean (DJF) when the NAO
amplitude is large (e.g. Hurrell and Deser, 2009), but it is also addressed in other seasons. Emphasis is given over
Canada, Europe and the North Atlantic Ocean in winter. The principal component (PC)-based NAO index (DJF)
provided by the Climate Analysis Section, NCAR, Boulder, USA (available at:
https://climatedataguide.ucar.edu/climate-data/hurrell-north-atlantic-oscillation-nao-index-pc-based) (last access: 15
June 2018) was used. Total ozone variability is also related to dynamical variability, for example variability in
tropopause height (e.g. Dameris et al., 1995; Hoinka et al., 1996; Steinbrecht et al., 1998). The impact of tropopause
height variations on total ozone variability was examined by analyzing the tropopause pressure from the
independently produced NCEP/NCAR (National Centers for Environmental Prediction/National Center for
Atmospheric Research) reanalysis 1 data set computed on a $2.5^{o}$ grid. The NCEP/NCAR reanalysis data were
provided from the web site at https://www.esrl.noaa.gov/psd/data/gridded/data.ncep.reanalysis.tropopause.html (last
access: 15 June 2018) (Kalnay et al., 1996).
**3 Results and discussion**
**3.1 Monthly zonal means and annual cycle**
Figure 1 compares monthly mean total ozone from GOME-2A and SBUV (v8.6) satellite overpass data for stations
shown in Table S1 (Supplement). The GOME-2A data were taken at a spatial resolution of $1^{o}$x$1^{o}$ around each of the
ground-based monitoring stations listed in Supplement Table S1 and then averaged over the tropics, middle and high
latitudes of both Hemispheres in $30^{o}$ latitudinal zones to provide the large scale monthly zonal means for the
GOME-2A data. Accordingly, SBUV satellite overpass data were averaged over each geographical zone to provide
the large scale zonal means for the SBUV observations. Mean differences and standard deviations between GOME-

218 2A and SBUV total ozone were found to be +0.1 ± 0.7% in the tropics (0-30 deg.), about +0.8 ± 1.6% in mid-
219 latitudes (30-60 deg.), about +1.3 ± 2.2% over the northern high latitudes (60-80 deg. N) and about -0.5 ± 2.9% over
220 the southern high latitudes (60-80 deg. S). The differences were estimated as [GOME-2A – SBUV] / SBUV (%)
221 from January 2007 to December 2016. Small differences were also found between GOME-2A and GB
222 measurements (Figure 2 and Table 1), where here GB stations data have been averaged over each geographical zone
223 to provide the large scale zonal means for the GB measurements. Mean differences and standard deviations between
224 GOME-2A and GB total ozone were found to be -0.7 ± 1.4% in the tropics (0-30 deg.), +0.1 ± 2.1% in mid-latitudes
225 (30-60 deg.), +2.5 ± 3.2% over the northern high latitudes (60-80 deg. N) and 0.0 ± 4.3% over the southern high
226 latitudes (60-80 deg. S). Recall that all estimates refer to the period between January 2007 and December 2016.

227 In summary, the largest differences between GOME-2A, SBUV (v8.6) and GB measurements are found over the
228 northern high latitudes (60$^{\circ}$-80$^{\circ}$ N) and the highest variability (standard deviation of the mean difference) is
229 observed over the latitude belt (60$^{\circ}$-80$^{\circ}$ S). In addition, these differences (especially at the high latitudes) can be
230 affected by the fact that the same days have not always been used for the construction of the monthly mean values
231 for the different datasets. In the tropics and mid-latitudes the respective differences are within ±1% or less, in line
232 with Chiou et al. (2014). Validation results were also presented by Loyola et al. (2011), Koukouli et al. (2012),
233 Coldewey-Egbers et al. (2015), Koukouli et al. (2015), updates of which are included in Hassinen et al. (2016). Our
234 results based on data updated to 2017 largely confirm those studies, pointing to the good performance of GOME-2A
235 when extending the period of record.

236 Next, we have studied the correlation between total ozone from GOME-2A and SBUV satellite data using linear
237 regression analysis for the period 2007–2016. The statistical significance of the correlation coefficients, $R$, was
238 calculated using the $t$-test formula for $R$ with $N$-2 degrees of freedom, as used in Zerefos et al. (2018). The
239 regression model showed statistically significant correlations between the different datasets as follows: $R$ = +0.99 in
240 the tropics, mid-latitudes and the northern high latitudes and $R$ = +0.97 in the southern high latitudes. All correlation
241 coefficients are highly statically significant (99.9% confidence level). In the long-term, statistically significant
242 correlation coefficients ($R \geq$ +0.94) are also found between GOME-2A satellite and GB measurements (Figure 2)
243 despite the different type of instruments used to measure total ozone from the ground. The regression parameters for
244 the correlation coefficients shown in Figures 1 and 2 are provided in Table 2.

245 A large part of the strong correlations shown in Figures 1 and 2 is attributable to the seasonal variability of total
246 ozone which is presented in Figure 3 for GOME-2A, SBUV and GB data. More specifically, Figure 3 shows the
247 seasonal variations of total ozone from station data, averaged per 10 degree latitude zones north and south. At high
248 latitudes our analysis stops at 80 degrees. There is a very good agreement between the annual cycles of total ozone
249 from the three datasets denoting the consistency of the satellite retrievals with GB observations. Similar annual
250 cycles are also found with the GTO-ECV ozone data (not shown). Similar consistency is also revealed for the
251 amplitudes of the annual cycles, computed as [(maximum value – minimum value)/2] in Dobson Units (DU). Figure
252 4 shows global maps of the amplitude of annual cycle of total ozone for the period 2007-2016 from GOME-2A
253 (upper left panel), GTO-ECV (upper right) and the TOMS/OMI/OMPS (lower left) satellite data. All maps are

plotted against the sine of latitude north and south in order to show areas according to their actual size. As can be
seen from Figure 4, the amplitude of annual cycle is less than 20 DU in the tropics, increasing as we move towards
middle and high latitudes up to 75 DU. Interestingly, there is a region with small amplitude of annual cycle in the
southern mid-latitudes with values of about 10-15 DU, seen in Figure 4 as a blue curved line crossing the longitudes
around 60 degrees south, which points to small seasonal variations of total ozone in these parts. The seasonal
increase in Antarctic ozone is delayed by 2-3 months compared to the north polar region. Only with the breakdown
of the polar vortex in late spring, i.e. at a time when the poleward transport over lower latitudes has already ceased,
does a strong ozone influx occur in the Antarctic. With this delay the amplitude of the seasonal variation stays much
smaller poleward of 55-60$^{o}$ in the south than in the north (Dütsch, 1974). These features are consistent between all
examined satellite data sets and are reproduced to a large extend by the Oslo CTM3 model as well, except in the
southern mid-latitudes where the model seems to underestimate the observed annual cycle (Figure 4 lower right).
In summary, we find a similar pattern and amplitude of annual cycle between total ozone from GOME-2A and the
other examined total ozone data sets. The mean differences in the annual cycles of GOME-2A and SBUV satellite
data are small in the tropics (0-30 deg.: 0.3 ± 2.4 DU), and increase as we move to mid-latitudes (30-60 deg.: 2.4 ±
4.4 DU) and higher latitudes (60-80 deg.: 1.7 ± 4.8 DU). These numbers are consistent with the ones found between
GOME-2A and GB measurements (tropics: 1.1 ± 2.3 DU; mid-latitudes: 1.2 ± 5.1 DU; high latitudes: 5.1 ± 7.1 DU).
In all latitude zones the correlation coefficients between the annual cycles of GOME-2A – SBUV and GOME-2A –
GB data pairs were found to be greater than 0.9.
Before examining correlations with the large scale natural fluctuations QBO, ENSO and NAO, the mean annual
cycle has been removed from the ozone data sets as described in the next section.
**3.2 Correlation with QBO**
We then studied how changes in dynamics affect the ozone columns in the atmosphere. The time series obtained
have been deseasonalised by subtracting the long-term monthly mean from each individual monthly mean value.
Ozone column variations for different latitude zones in the Northern and Southern Hemispheres have been
compared. Figure 5 compares total ozone deseasonalised anomalies (in % of the mean) from GOME-2A and SBUV
satellite retrievals in the tropics (10$^{o}$ N–10$^{o}$ S), sub-tropics (10$^{o}$–30$^{o}$) and mid-latitudes (30$^{o}$–60$^{o}$). The right panel of
Figure 5 shows the respective anomalies from GTO-ECV data. Mean differences between GOME-2A and SBUV
deseasonalised monthly zonal means between 60$^{o}$ N and 60$^{o}$ S are less than ±0.5%.
The line with dots superimposed on the ozone anomalies in Figure 5 shows the equatorial zonal winds at 30 hPa
which were used as a proxy index to study the impact of QBO on total ozone. The general features include a QBO
signal in total ozone at latitudes between 10$^{o}$ N and 10$^{o}$ S which almost matches with the phase of QBO in the zonal
winds. At higher northern and southern latitudes there is a phase shift in the QBO impact on total ozone. The impact
of QBO is most pronounced in the tropics and less pronounced in the sub-tropics and mid-latitudes. Strong positive
correlations with the QBO are found in the tropics (correlation between GOME-2A and QBO of about +0.77, t-test
= 12.91) and weaker (usually of opposite sign) less significant correlations are found at higher latitudes (about −0.15
in the northern and about −0.45 in the southern extra tropics). Similar correlation patterns with the QBO are found
for the GTO-ECV, SBUV and GB data. These correlations suggest that the variability that can be attributed to the
QBO in the tropics is about 60%, and about 2% and 20% in the northern and the southern extra tropics, respectively.
Table 3 summarizes the correlation and regression coefficients between total ozone and QBO at 30 hPa for the
different latitude zones and the different datasets. For latitudes between $10^o$ N and $10^o$ S correlations between total
ozone from GOME-2A, GTO-ECV, SBUV, GB data and the QBO are all positive. At latitudes between $10^o$ and $30^o$
the correlations turn to negative, in agreement with Knibbe et al. (2014) results, who noted that moving from the
tropics towards higher latitudes the regression estimates switch to negative values at approximately $10^o$ N and $10^o$ S.
The correlations with the QBO at 30 hPa remain negative up to $60^o$, a consistent result among all our data sets,
something also reported by Knibbe et al. (2014) with the MSR ozone data. The correlation and regression
coefficients between GOME-2A and QBO are fairly similar to those found between SBUV and QBO, as well as
among all data sets as seen in Table 3, despite the different periods of records.
These features are also evident in Figure 6 which compares GOME-2A (and GTO-ECV) satellite total ozone with
GB observations with respect to the QBO. Mean differences and standard deviations between GOME-2A and GB
and between GTO-ECV and GB deseasonalised total ozone data do not exceed one percent. Again, correlation
coefficients between deseasonalised GOME-2A and deseasonalised GB data are highly significant in all latitude
zones ($30^o$–$60^o$ N: +0.91 (slope=0.818, error=0.035, t-value=23.466, N=119); $10^o$–$30^o$ N: +0.91 (slope=0.786,
error=0.033, t-value=23.529, N=119; $10^o$ N–$10^o$ S: +0.94 (slope=0.973, error=0.034, t-value=28.449, N=109; $10^o$–
$30^o$ S: +0.87 (slope=0.864, error=0.044, t-value=19.659, N=119; $30^o$–$60^o$ S: +0.88 (slope=0.858, error=0.043, t-
value=19.854, N=119). The same stands for the correlations between GTO-ECV and GB data pairs ($30^o$–$60^o$ N:
+0.94; $10^o$–$30^o$ N: +0.89; $10^o$ N–$10^o$ S: +0.94; $10^o$–$30^o$ S: +0.87; $30^o$–$60^o$ S: +0.85). Our results are in line with
Eleftheratos et al. (2013) and Isaksen et al. (2014) who compared QBO-related ozone column variations from the
chemical transport model Oslo CTM2 with SBUV satellite data for shorter time periods. In summary, it has been
shown that GOME-2A depicts the significant effects of QBO on stratospheric ozone in concurrence with SBUV and
GB measurements. The instrument captures correctly the variability of ozone in the tropics and the mid-latitudes,
which is nearly in phase with the QBO in the tropics and out of phase in the northern and the southern mid-latitudes
as have been shown by earlier studies (e.g. Zerefos, 1983; Baldwin et al., 2001).
**3.3 Correlation with ENSO**
Apart from the QBO, which affects the variability of total ozone in the tropics, an important mode of natural climate
variability in the tropics is ENSO. To examine the impact of ENSO on total ozone in the tropics we first removed
variability related to the QBO and the solar cycle, and then performed the correlation analysis with the SOI. The
effect of the QBO was removed from the time series by using a linear regression model for the total ozone variations
at each grid box, of the form:
$D(t) = a0 + a1 * QBO(t) + residuals(t); 0 < t \leq T$ (1)

where D($t$) is the monthly deseasonalised total ozone and $t$ is the time in months with $t$=0 corresponding to the initial month and $t$=$T$ corresponding to the last month. The term $a0$ is the intercept of the statistical model. To model QBO we made use of the equatorial zonal winds at 30 hPa. The term $a1$ is the regression coefficient of QBO. The QBO component was removed from the time series by using a phase lag with maximum correlation of 28 months (month lag -14 to month lag 13). The QBO-related coefficients $\alpha0$ and $\alpha1$ of Eq. (1) for the deseasonalized GOME-2A, GTO-ECV, TOMS/OMI/OMPS and Oslo CTM3 zonal mean data are presented in Table 3. Additional information for the regression coefficients $\alpha1$ of QBO is provided in the Supplement Figure S1, which shows the spatial distribution of the regression coefficients in latitude-longitude maps.

The residuals from Eq. (1) were then inserted in a second regression (Eq. 2) to account for the effect of solar cycle on total ozone, as follows:

$$O_3(t) = \beta0 + \beta1 * F10.7(t) + residuals(t); 0 < t \leq T \qquad (2)$$

where $\beta0$ and $\beta1$ are now the intercept and regression coefficients of solar cycle, respectively. To model the solar cycle we used the 10.7 cm wavelength solar radio flux (F10.7) as a proxy, taken from the National Research Council and Natural Resources Canada at ftp://ftp.geolab.nrcan.gc.ca/data/solar_flux/monthly_averages/solflux_monthly_average.txt (last access 12 December 2018). We use the absolute solar fluxes, which are adjusted to account for variations in Earth-Sun distance and uncertainty in antenna gain and waves reflected from the ground. Latitude-longitude maps of the regression coefficients $\beta1$ of the solar cycle are presented in the Supplement Figure S2. We note that the global pattern of the regression coefficients of solar cycle from GOME-2A data matches well with what has been shown by Knibbe et al. (2014) with the reanalysis MSR data.

The remainders from Eq. (2) were used in a third regression (Eq. 3) to study the correlations between total ozone and SOI at each individual grid box:

$$O_3(t) = c0 + c1 * SOI(t) + residuals(t); 0 < t \leq T \qquad (3)$$

where $c0$ and $c1$ are now the intercept and regression coefficients of ENSO, accordingly. Estimates of the regression coefficients $c1$ are shown in the Supplement Figure S3.

Figure 7 presents the correlations between SOI and total ozone from GOME-2A (upper left panel), GTO-ECV (upper right) and TOMS/OMI/OMPS satellite data (bottom left), as well as between SOI and the Oslo model simulations (bottom right). All four plots refer to the period 2007-2016. As can be seen from Figure 7 (upper left), correlations of >0.3 between GOME-2A total ozone and SOI are found in the tropical Pacific Ocean at latitudes between 25 deg. north and south. These correlations were tested as to their statistical significance in the period 2007-2016 using the $t$-test for $R$ with $N$-2 degrees of freedom (as in Zerefos et al., 2018), and were found to be statistical significant. A similar picture of correlation coefficients is also observed by the GTO-ECV and TOMS/OMI/OMPS data. Both data sets show similar results as to the range of correlations (>0.3) in the tropical Pacific for the common period of observations. Nevertheless, the spatial resolution is higher in the GOME-2A and

GTO-ECV (1x1 deg.) data than in the TOMS/OMI/OMPS (5x5 deg.) data, so the former data sets perform better
when looking at smaller space scales. We have to note here that in both maps there are larger areas with correlation
coefficients >0.3 in the southern part of the tropics than in the northern part. However, this was mostly observed
during the period 2007-2016. By examining the longer-term data record of the TOMS/OMI/OMPS data which
extend back to 1979, we find symmetry in the pattern of correlations north and south of the equator in the tropical
Pacific Ocean (Figure A1 of Appendix A), which indicates that both sides of the tropical Pacific are affected more
or less in a similar way by El Niño/La Niña events. Finally, the Oslo CTM3 gives small correlations (<0.3) in the
tropical Pacific Ocean around the equator, except over the northern and southern subtropics where the model
compares better with the observations.
The small rectangle in Figure 7 corresponds to the South Pacific region ($10^{o}$-$20^{o}$ S, $180^{o}$-$220^{o}$ E) and the blue cross
to the station Samoa (American Samoa; $14.25^{o}$ S, $189.4^{o}$ E), at which total ozone has been studied as for the impact
of ENSO after removing variability related to the annual cycle, QBO and the solar cycle. Figure 8 shows an example
of the ENSO impact on total ozone in the South Pacific Ocean. The upper panel shows the time series of total ozone
anomalies from GOME-2A, GTO-ECV and TOMS/OMI/OMPS satellite data together with the SOI (Figure 8a).
Comparisons of GOME-2A data with GTO-ECV data, SBUV overpass data and GB measurements at the station
Samoa are shown in Figure 8b. The dotted line shows the respective tropopause pressure anomalies from NCEP
reanalysis. All data sets point to the strong influence of ENSO on total ozone. Most evident is the strong decrease of
about 4% in 1997/98 which was caused by the strongest El Niño event in the examined period. A strong decrease is
also observed in the tropopause pressures by NCEP. Notable also is the strong La Niña event in 2010 which caused
total ozone to increase by about 4%. We calculate a strong correlation between total ozone from GTO-ECV and SOI
of +0.66 (99% confidence level), which accounts for about 40% of the variability of total ozone over the tropical
Pacific Ocean when the annual cycle, QBO signal and solar cycle are removed. From the regression with SOI we
estimated an ENSO-related term from which we calculated the amplitude of ENSO in total ozone as [maximum
ozone - minimum ozone]/2. The amplitude of ENSO in total ozone was estimated to be 8.8 DU or 3.5% of the
annual mean. This is comparable to the amplitude of annual cycle (7.7 DU or 3.0% of the mean) and larger than the
amplitude of QBO (2.2 DU or 0.8% of the mean) and the amplitude of solar cycle in this region (4.1 DU or 1.6% of
the mean). These results are based on the GTO-ECV total ozone data. Similar results were also found at the station
Samoa from GB observations (i.e. correlation with SOI: +0.55, amplitude of ENSO: 7.7 DU or 3.0% of the mean,
amplitude of annual cycle: 6.7 DU or 2.7% of the mean). Statistics of total ozone such as mean, amplitude of annual
cycle, amplitude of QBO, amplitude of solar cycle and amplitude of ENSO in total ozone over the selected areas are
presented in Table 4. Both satellite, GB and model data show consistent results. It also appears that the station
Samoa represents well the greater area in the Southern Pacific as to the impact of ENSO.
Differences between GOME-2A and its data pairs in the southern Pacific Ocean are the order of -0.2 ± 1.0%
between GOME-2A and TOMS/OMI/OMPS data, -0.3 ± 0.9% between GOME-2A and GTO-ECV, and -0.9 ± 1.8%
between GOME-2A and Oslo CTM3. Accordingly, differences at Samoa are: -0.6 ± 1.9% between GOME-2A and
GB data, 0.0 ± 1.4% between GOME-2A and GTO-ECV, and -0.1 ± 1.3% between GOME-2A and SBUV. Despite
the small differences found, we note here that GOME-2A values in the last 4 years of Figures 8 and 9 slightly
deviate from the other data sets, and correlate weaker with SOI than the other years in the time series. For instance,
we estimate a drop in the correlation coefficient between GOME-2A and SOI at the station Samoa (+0.58 in the
period 2007-2012 and +0.47 in the period 2007-2016), which nevertheless does not alter the statistical significance
of the correlation.
From Figure 8 it also appears that there are high correlations with the tropopause height. The correlation coefficient
between the NCEP tropopause pressure and GOME-2A total ozone over the South Pacific Ocean is of the order of
+0.59 (Student's t-test statistics results: t-value = 7.946, p-value <0.0001, N = 119). Accordingly, the correlation
with GTO-ECV ozone data is the order of +0.64 (t-value = 13.165, p-value <0.0001, N = 252) and with
TOMS/OMI/OMPS the order of +0.58 (t-value = 10.913, p-value <0.0001, N = 241). The high correlation between
the tropopause pressure and total ozone on interannual and longer time scales points to the very strong link between
these parameters. These links were already documented in the past (e.g. Steinbrecht et al., 1998, 2001) and are
verified with the GOME-2A data. At the same time a strong correlation is also evident between tropopause pressure
and SOI, again on interannual and longer time scales ($R$= +0.66, t-value = 13.825, p-value <0.0001, N = 252). The
above results point to the strong impact of ENSO on the tropical ozone column through the tropical tropopause;
warm (El Niño) and cold (La Niña) events affect the variability of the tropopause which in turn affects the
distribution of stratospheric ozone. In the tropics, where total ozone is mainly stratospheric, as the tropopause moves
to higher altitudes (lower pressure), the stratosphere is compressed, reducing the amount of stratospheric (total)
ozone. This happens during warm (El Niño) episodes. The opposite phenomenon occurs during cold (La Niña)
events when the tropopause height decreases (higher pressure) and total ozone is then increased. These events can
affect the long-term ozone trends in the tropics when looking at time periods when strong El Niño and La Niña
events occur at the beginning and the end of the trend period respectively (Coldewey-Egbers et al., 2014).
Furthermore, in Figure 8 we have marked 7 stations in the greater South Asia region (35°-45° N, 45°-125° E) where
total ozone is anti-correlated with the SOI. Admittedly, these anti-correlations are weak (about -0.3) but we thought
worthwhile presenting the time series in these areas as well. Figure 9 shows the variability of total ozone after
removing seasonal, QBO and solar cycle related variations, over the South Asia region (upper panel) and over the 7
stations averaged within this region (lower panel). As can be seen from this figure, the explained variance by ENSO
is small, not exceeding 9%. All correlations from the comparisons with the SOI are summarized in Table 5. In spite
the small correlations with the SOI, the consistency between GOME-2A, GTO-ECV, TOMS/OMI/OMPS and Oslo
CTM3 data anomalies is very high and their differences are within ± 1%. Differences at the 7 stations in South Asia
are as follows: -1.3 ± 2.4% between GOME-2A and GB data, -0.4 ± 1.0% between GOME-2A, and GTO-ECV and -
0.5 ± 1.0% between GOME-2A and SBUV.
In summary, our findings indicate that GOME-2A captures well the disturbances in total ozone during ENSO events
with respect to satellite SBUV and GB observations. Our findings on the ENSO-related total ozone variations (low
ozone during ENSO warm events, high ozone during ENSO cold events, and magnitude of changes) are in line with
recent studies (e.g. Randel and Thompson, 2011; Oman et al., 2013, Sioris et al., 2014) included in the 2014 Ozone
Assessment report (Pawson et al., 2014; WMO, 2014). Our results are also in agreement with Knibbe et al. (2014)
who showed negative ozone effects of El Niño between $25^{o}$ S and $25^{o}$ N, especially over the Pacific.
**3.4 Correlation with NAO**
The residuals from Eq. (3), free from seasonal, QBO, solar and ENSO related variations, were later used to study the
correlation between total ozone and NAO in winter. The results are presented in Figure 10 which shows the
correlation coefficients between total ozone and NAO index in winter from the GOME-2A (upper left), GTO-ECV
(upper right) and TOMS/OMI/OMPS satellite data (lower left), and the Oslo CTM3 model calculations (lower
right). Negative correlations between total ozone and NAO are presented with blue colours while positive
correlations with red colours. From Figure 10 (upper left) it appears that total ozone is strongly correlated with NAO
in many regions. Strong negative correlation coefficients are observed in the majority of the northern mid-latitudes
(R about −0.6) while positive correlations exist in the tropics and some negative correlations in the southern mid-
latitudes. These characteristics are observed in both GTO-ECV and TOMS/OMI/OMPS datasets and are reproduced
by the Oslo model as well, all for the common period 2007-2016. The regression coefficients on these comparisons
are presented in the Supplement Figure S4.
We note here that the results of the correlation analysis for the period 2007-2016 were based on a relative small
sample of data from 10 winters and therefore many of these correlation coefficients may not be statistically
significant. The statistical significance of the correlation coefficients in every grid box was tested only with the
TOMS/OMI/OMPS data (Figure A2, Appendix A), which provided us the opportunity to calculate the respective
correlations using data for the whole period of record 1979-2016. It appears that when extending the data back to the
1980's the negative correlations in the southern mid-latitudes in winter disappear while the positive correlations in
the tropics become weaker; yet the observed anti-correlation between total ozone and NAO index in the northern
mid-latitude zone holds strong. The dotted line in the plot shows areas with statistically significant correlation
coefficients (99% confidence level). Indeed, in the long-term, statistically significant correlations between total
ozone and the NAO index during winter are mostly found over the northern mid-latitudes and the sub-tropics. A
small, statistically significant signal is also seen over Antarctica but it was not analysed further.
According to this finding we have restricted the analysis of NAO to the northern mid-latitudes. Rectangles (Figure
10, upper left) correspond to two regions in the North Atlantic, i.e., $35^{o}$-$50^{o}$ N, $20^{o}$-$50^{o}$ W and $15^{o}$-$27^{o}$ N, $30^{o}$-$60^{o}$ W
respectively, which were studied for the impact of NAO on total ozone after removing variability related to the
annual cycle, QBO, solar cycle and ENSO. In addition we have studied a number of stations in Canada, USA, and
Europe contributing ozone data to WOUDC, which are marked by red and green crosses in Figure 10. The red
crosses refer to the monitoring stations in Canada and the US, and the green crosses to the stations in Europe. In
Figure 11 we present the times series of total ozone anomalies from GOME-2A, GTO-ECV and TOMS/OMI/OMPS
satellite data along with the NAO index in winter over the North Atlantic. Model calculations are shown as well.
The dotted line shows the respective tropopause pressure anomalies from NCEP reanalysis. Comparisons between
GOME-2A, GTO-ECV, SBUV (v8.6) overpass data and GB measurements over the various stations in Canada,
USA and Europe are shown in Figure 12.

The observed anomalies over the North Atlantic Ocean point to the strong influence of NAO on total ozone in winter. Most evident is the strong increase in total ozone in 2010 of more than 8% particularly over $35^{o}$-$50^{o}$ N and $20^{o}$-$50^{o}$ W. This increase was accompanied by a strong increase in tropopause pressures. Both changes (in total ozone and tropopause pressures) occurred under a strong negative phase of NAO, the strongest one in the past 20 years. We observe strong anti-correlation among total ozone and NAO index in winter ($R$= –0.74 over $35^{o}$-$50^{o}$ N, $20^{o}$-$50^{o}$ W), which is statistically significant at the 99% confidence level. This anti-correlation suggests that about 50% of the variability of total ozone in winter is explained by NAO when the annual cycle, QBO, solar cycle and ENSO signals are removed. Differences for GOME-2A and its data pairs are estimated to be -0.7 ± 1.1% between GOME-2A and TOMS/OMI/OMPS data, +0.1 ± 1.0% between GOME-2A and GTO-ECV, and -0.2 ± 1.5% between GOME-2A and Oslo CTM3. From the regression with the NAO index we derived a NAO-related term from which we calculated the amplitude of NAO in total ozone as [maximum ozone - minimum ozone]/2. The amplitude of NAO over the North Atlantic region ($35^{o}$-$50^{o}$ N, $20^{o}$-$50^{o}$ W) was estimated to be about 16.5 DU or 5.2% of the annual mean. This is about half of the amplitude of the annual cycle (which is ~37 DU or 11.7% of the mean). These estimates are based on GTO-ECV data. Similar correlation and amplitude were also found with GOME-2A, the combined TOMS/OMI/OMPS satellite data and the Oslo CTM3 model simulations.

A similar but opposite correlation is found over the southern part of the North Atlantic ($15^{o}$-$27^{o}$ N, $30^{o}$-$60^{o}$ W). Here, we estimate a significant correlation coefficient with NAO of +0.60, amplitude of NAO of about 7.2 DU (2.6% of the annual mean) and amplitude of annual cycle of about 15.8 DU (5.7% of the mean). Again, similar estimates are found with the GOME-2A and the TOMS/OMI/OMPS satellite data and reproduced by the model calculations as well. The annual mean total ozone and the amplitudes of annual cycle, QBO, solar cycle and NAO in total ozone over the studied regions in the North Atlantic are summarised in Table 6. Differences between GOME-2A and GTO-ECV data at the southern part of North Atlantic are the order of -0.6 ± 0.7%. Differences with the TOMS/OMI/OMPS data are estimated to be -0.9 ± 0.8%, and with the Oslo CTM3 -0.1 ± 0.7%.

The time series of total ozone anomalies and of the NAO index for the examined stations in Canada, USA and Europe are presented in Figure 12. Table 7 presents the respective statistics. The correlation between total ozone and the NAO index in winter after removing from ozone variability related to the annual cycle, QBO, solar cycle and ENSO is –0.40 (90% confidence level). Again, a particular feature was the total ozone increase in 2010 by 6% of the mean associated with the negative NAO phase. Noteworthy on this increase is the consistency with the GB measurements and the satellite SBUV overpass data, and in general the agreement found between the variability of the tropopause pressures and total ozone. Differences between GOME-2A and GB data are -1.0 ± 1.8%. Accordingly we estimate differences of about -1.1 ± 0.5% between GOME-2A and GTO-ECV data and of about -1.3 ± 0.6% between GOME-2A and SBUV data. On the basis of GTO-ECV data we estimate that in Canada and USA, the amplitude of NAO in total ozone in winter is about 7 DU (or 2.2% of the mean), while it is estimated to be about 9 DU (or 2.7% of the mean) over Europe. These numbers are slightly smaller than the GOME-2A, GB and SBUV estimates, less than about one percent (Table 7).

The anti-correlation between total ozone column and the NAO index during winter also applies to southern Europe
and the Mediterranean. Following the study of Ossó et al. (2011) who reported a reversal in the correlation pattern
between NAO and total ozone from winter to summer in southern Europe, we have looked at the correlations during
summer as well. Figure 13 presents the comparisons for 21 ground-based stations located in the region bounded by
latitudes $30^o$-$47^o$ N and by longitudes $10^o$W-$40^o$E. Figure 13a shows results for the summer and Figure 13b shows
results for winter. As evident, the observed anti-correlation between GB total ozone and NAO in winter (R= -0.43,
slope= -0.980, t-value= -2.095, p-value= 0.0499, N = 21) reverses sign and becomes positive in the summer (R=
+0.60, slope= 0.874, t-value= 3.309, p-value= 0.0037, N= 21), indicating that the NAO explains about 36% of ozone
variability in the summer in this region. A similar picture is also seen from GOME-2A, GTO-ECV and SBUV data.
In summary, our findings based on GOME-2A, GTO-ECV and SBUV overpass data are in line with those found by
Ossó et al. (2011) and Steinbrecht et al. (2011) who analysed TOMS and OMI satellite data, and GB measurements
at the station Hohenpeissenberg, respectively. During winter, total ozone variability associated with the NAO is
particularly important over northern Europe, the U.S. East Coast, and Canada, explaining up to 30% in total ozone
variance for this region (Ossó et al., 2011). Also, both studies found unusually high total ozone columns in 2010
over much of the Northern Hemisphere and related them to the negative phase of NAO or AO (the Arctic
Oscillation).
**4 Conclusions**
We have evaluated the ability of GOME-2/MetopA (GOME-2A) satellite total ozone retrievals to capture known
natural oscillations such as the QBO, ENSO and NAO. In general, GOME-2A depicts these natural oscillations in
concurrence with GTO-ECV, TOMS/OMI/OMPS, SBUV (v8.6) satellite overpass data, ground-based
measurements (Brewer, Dobson, filter and SAOZ) and chemical transport model calculations (Oslo CTM3).
Mean differences between GOME-2A and SBUV total ozone were found to be +0.1 ± 0.7% in the tropics (0-30
deg.), about +0.8 ± 1.6% in mid-latitudes (30-60 deg.), about +1.3 ± 2.2% over the northern high latitudes (60-80
deg. N) and about -0.5 ± 2.9% over the southern high latitudes (60-80 deg. S). These differences were estimated as
[GOME-2A – SBUV] / SBUV (%) from January 2007 to December 2016. Small differences were also found
between GOME-2A and GB measurements, with standard deviations of the differences being ± 1.4% in the tropics,
± 2.1% in mid-latitudes, and ± 3.2% and ± 4.3% over the northern and the southern high latitudes respectively.
The variability of total ozone from GOME-2A has been compared with the variability of total ozone from other
examined data sets as to their agreement depicting natural atmospheric phenomena such as the QBO, ENSO and
NAO. First, we studied correlations between total ozone and the QBO after removing from the ozone data sets
variability related to the seasonal cycle. Then, we examined correlations between total ozone and ENSO, after
removing variability related to the QBO and solar cycle, and finally correlations with the NAO after removing
variability related to the QBO, solar cycle and ENSO. Our main results are as follows:
**QBO**: Total ozone from GOME-2A is well correlated with the Quasi-Biennial Oscillation (+0.8 in the tropics) in
agreement with GTO-ECV, SBUV and GB data. The amplitude of QBO on total ozone maximizes around the
equator and it is estimated to about 2.6% of the mean. Going from low to mid-latitudes there is a phase shift in the
QBO impact on total ozone. Correlation coefficients between GOME-2A total ozone and the QBO over 30-60 deg.
north and south are -0.1 and -0.5 respectively, in agreement with the correlations between GB total ozone and the
QBO (-0.2 and -0.5, accordingly). On the basis of GOME-2A, the amplitude of QBO in total ozone is estimated to
be 0.6% of the mean in the northern mid-latitudes and 1.4% of the mean in the southern mid-latitudes.
**ENSO**: Correlation coefficients among GOME-2A total ozone and SOI in the tropical Pacific Ocean are estimated
to be about +0.6, consistent with GTO-ECV, SBUV and GB observations. It was found that the El Nino Southern
Oscillation (ENSO) signal is evident and consistent in all examined datasets. The amplitude of ENSO in total ozone
is about 6–9 DU corresponding to about 2.5–3.5% of the annual mean. Differences between GOME-2A, GTO-ECV
and GB measurements during warm (El Niño) and cold (La Niña) events are within ±1.5%. Similar estimates also
result from the Dobson measurements at American Samoa, indicating that Samoa station represents well the greater
area in the Southern Pacific for satellite evaluations as to the impact of ENSO.
**NAO**: The respective results as far as the impact of the North Atlantic Oscillation over the northern mid-latitudes
showed a clear NAO signal in winter in all data sets, with amplitudes of about 16-19 DU (about 5–6% of the annual
mean) in the North Atlantic, 9-12 DU (3-4% of the mean) over Europe, and 7-10 DU (2-3% of the mean) over
Canada and the US. Comparison with GB observations over Canada and Europe showed very good agreement
between GOME-2A, GTO-ECV and GB observations as to the influence by NAO, with differences within ±1%.
In addition to the usual validation methods, which compare monthly mean and zonal mean total ozone data and
analyse the differences between satellite and GB instruments, we showed here that quasi cyclical perturbations such
as the QBO, ENSO and NAO can serve as independent proxies of spatiotemporal variation to qualitatively evaluate
GOME-2A satellite total ozone against ground-based and other satellite total ozone data sets. The agreement and
small differences which were found between the variability of total ozone from GOME-2A and the variability of
total ozone from other satellite retrievals and ground-based measurements during these naturally-occurring
oscillations verify the good quality of GOME-2A satellite total ozone to be used in ozone-climate research studies.
**Data availability**
Satellite  SBUV  (v8.6)  total  ozone  station  overpass  data  were  downloaded  from  https://acd-
ext.gsfc.nasa.gov/Data_services/merged/index.html (last access: 15 June 2018) (McPeters et al., 2013; Bhartia et al.,
2013). GTO-ECV total ozone data are available at http://www.esa-ozone-cci.org/?q=node/160 (last access: 15 June
2018) (Coldewey-Egbers et al., 2015; Garane et al., 2018). Ground-based total ozone daily summaries were obtained
from  the  World  Ozone  and  UV  Data  Centre  (WOUDC)  at
https://woudc.org/archive/Summaries/TotalOzone/Daily_Summary/ (last access: 15 June 2018). The QBO
component on total ozone was examined by using the monthly mean zonal winds at Singapore at 30 hPa. Zonal

wind data at 30 hPa were provided by the Freie Universität Berlin (FU-Berlin) at http://www.geo.fu-berlin.de/met/ag/strat/produkte/qbo/qbo.dat (last access: 15 June 2018) (Naujokat, 1986). The Southern Oscillation Index (SOI) was provided by the Bureau of Meteorology of the Australian Government at http://www.bom.gov.au/climate/current/soi2.shtml (Australian Government – Bureau of Meteorology, 2018). The NAO index for December, January and February was provided by the Climate Analysis Section, NCAR, Boulder, USA at https://climatedataguide.ucar.edu/climate-data/hurrell-north-atlantic-oscillation-nao-index-pc-based (last access: 15 June 2018) (Hurrell and Deser, 2009). The tropopause pressures from the NCEP/NCAR reanalysis 1 data set were downloaded from https://www.esrl.noaa.gov/psd/data/gridded/data.ncep.reanalysis.tropopause.html (last access: 15 June 2018) (Kalnay et al., 1996).

**Competing interests**

The authors declare that they have no conflict of interest.

**Acknowledgements**

Development of the GOME-2/MetopA total ozone products and their validation has been funded by the AC SAF project with EUMETSAT and national contributions. We acknowledge support of this work by the project "PANhellenic infrastructure for Atmospheric Composition and climatE change" (MIS 5021516) which is implemented under the Action "Reinforcement of the Research and Innovation Infrastructure", funded by the Operational Programme "Competitiveness, Entrepreneurship and Innovation" (NSRF 2014-2020) and co-financed by Greece and the European Union (European Regional Development Fund). We further acknowledge the Mariolopoulos-Kanaginis Foundation for the Environmental Sciences, the ESA Ozone CCI project and the NASA Goddard Space Flight Centre. The ground-based data used in this publication were obtained as part of WMO's Global Atmosphere Watch (GAW) and are publicly available via the World Ozone and UV Data Centre (WOUDC). The authors would like to thank all the investigators that provide quality assured total ozone column data on a timely basis to the WOUDC database. We acknowledge the National Oceanic and Atmospheric Administration (NOAA) for maintaining the American Samoa Dobson station. KE and CS would like to dedicate the study to the memory of Professor Ivar Isaksen (University of Oslo) who passed away on May 16[th], 2017.

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

764

**Appendix A**

**Correlation with SOI (month lag 0)**
**combined TOMS/OMI/OMPS 5x5 (1979-2016)**

**Figure A1. Map of correlation coefficients between total ozone from TOMS/OMI/OMPS satellite data and SOI for the whole period 1979-2016, after removing variability related to the seasonal cycle, QBO and solar cycle. The dotted line bounds the regions where the correlation coefficients are statistically significant at the 99% confidence level (t-test). Only correlation coefficients above/below ±0.2 are shown. Ozone data for the period 1991-1993 after the Mt Pinatubo eruption were not used in the correlation analysis to avoid any data contamination by the volcanic aerosols.**

777

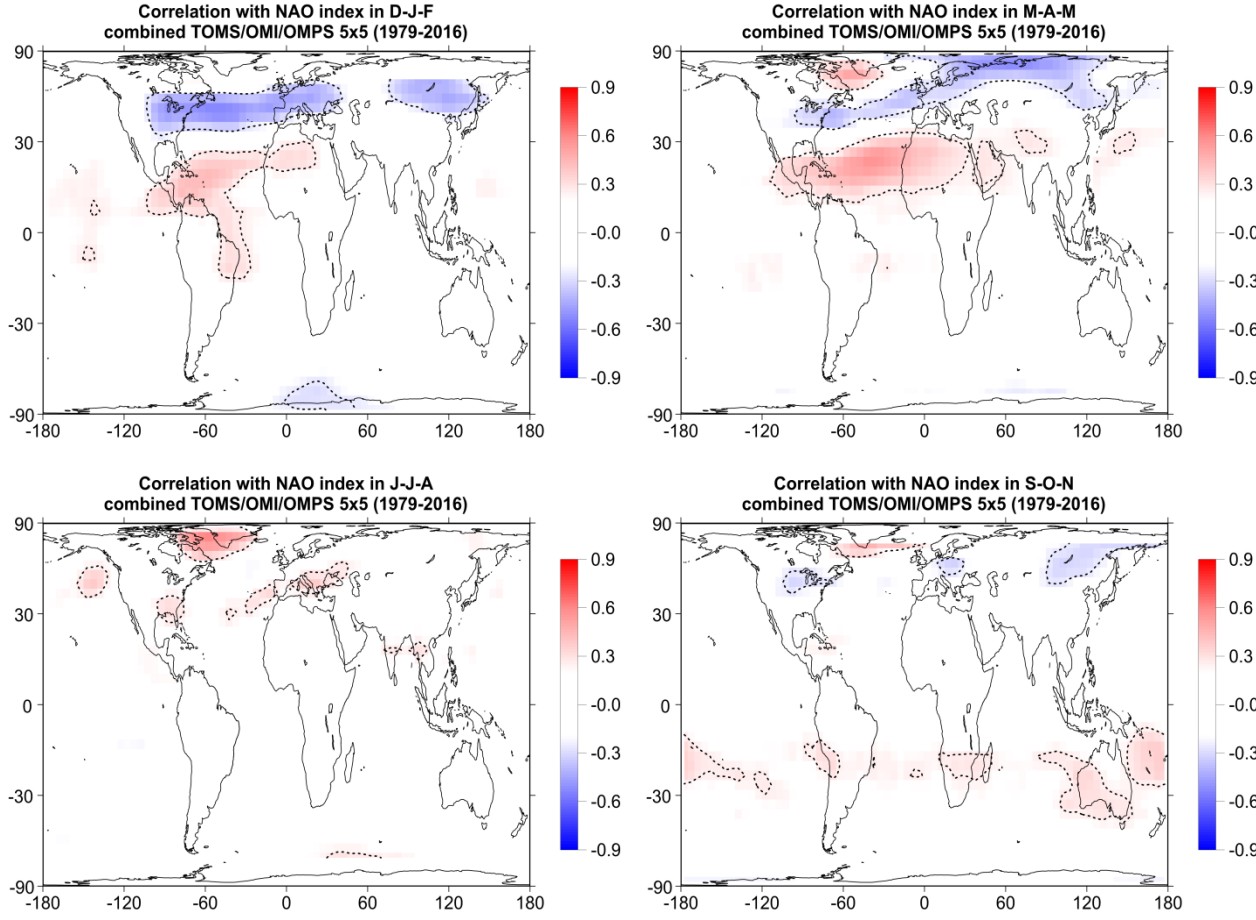

778

**Figure A2. Map of correlation coefficients between total ozone from TOMS/OMI/OMPS satellite data and the NAO index during winter (December, January, February (D-J-F); upper left), spring (March, April, May (M-A-M); upper right), summer (June, July, August (J-J-A); lower left) and autumn (September, October, November (S-O-N); lower right) for the whole period 1979-2016, after removing variability related to the seasonal cycle, QBO, solar cycle and ENSO. The dotted line bounds the regions where the correlation coefficients are statistically significant at the 99% confidence level (t-test). Only correlation coefficients above/below ±0.2 are shown. Ozone data for the period 1991-1993 after the Mt Pinatubo eruption were not used in the correlation analysis to avoid any data contamination by the volcanic aerosols.**



**Table 1. Mean differences and their standard deviations in percent between total ozone from GOME-2A, SBUV (v8.6) satellite overpass data and ground-based observations over different latitude zones, as shown in Figures 1 and 2.**


| | [GOME-2A – SBUV] / SBUV (%) Stations mean data | [GOME-2A – GROUND] / GROUND (%) Stations mean data |
|---|---|---|
| 60°-80° N | +1.3 ± 2.2 | +2.5 ± 3.2 |
| 30°-60° N | +0.8 ± 1.6 | +0.1 ± 1.9 |
| 0°-30° N | 0.0 ± 0.7 | −0.5 ± 1.2 |
| 0°-30° S | +0.1 ± 0.7 | −0.9 ± 1.6 |
| 30°-60° S | +0.9 ± 1.6 | 0.0 ± 2.4 |
| 60°-80° S | −0.5 ± 2.9 | 0.0 ± 4.3 |



**Table 2. Statistics of the correlations shown in Figures 1 and 2 between total ozone from a) GOME-2A data and SBUV (v8.6) overpass data, and b) GOME-2A data and ground-based measurements.**


| (a) GOME-2A and SBUV (v8.6) | Correlation | Intercept (DU) | Slope* | Error | t-value | p-value | N |
|---|---|---|---|---|---|---|---|
| 60°-80° N | +0.987 | 4.925 | 0.999 | 0.015 | 65.224 | <0.0001 | 117 |
| 30°-60° N | +0.984 | 5.002 | 0.993 | 0.017 | 59.784 | <0.0001 | 118 |
| 0°-30° N | +0.989 | 28.304 | 0.894 | 0.012 | 72.404 | <0.0001 | 118 |
| 0°-30° S | +0.981 | 21.575 | 0.919 | 0.017 | 53.874 | <0.0001 | 118 |
| 30°-60° S | +0.977 | –4.198 | 1.023 | 0.021 | 49.123 | <0.0001 | 118 |
| 60°-80° S | +0.974 | 2.944 | 0.984 | 0.025 | 39.985 | <0.0001 | 88 |


| (b) GOME-2A and Ground-based | Correlation | Intercept (DU) | Slope* | Error | t-value | p-value | N |
|---|---|---|---|---|---|---|---|
| 60°-80° N | +0.973 | 7.651 | 1.002 | 0.022 | 45.155 | <0.0001 | 118 |
| 30°-60° N | +0.977 | 15.772 | 0.952 | 0.019 | 49.671 | <0.0001 | 119 |
| 0°-30° N | +0.982 | 49.534 | 0.810 | 0.014 | 56.951 | <0.0001 | 119 |
| 0°-30° S | +0.916 | 56.520 | 0.778 | 0.032 | 24.655 | <0.0001 | 119 |
| 30°-60° S | +0.946 | 12.423 | 0.958 | 0.030 | 31.612 | <0.0001 | 119 |
| 60°-80° S | +0.939 | 0.405 | 0.999 | 0.039 | 25.439 | <0.0001 | 89 |

* Error, t-value and p-value refer to slope.


**Table 3. Statistics of correlations between deseasonalized total ozone and the QBO at 30 hPa for a) GOME-2A data, b) GTO-ECV data, c) SBUV (v8.6) overpass data, and d) ground-based measurements.**


| (a) GOME-2A and QBO | Correlation | Intercept (%) | Slope* | Error | t-value | p-value | N |
|---|---|---|---|---|---|---|---|
| 30º-60º N | –0.073 | –0.045 | –0.008 | 0.010 | –0.791 | 0.4307 | 119 |
| 10º-30º N | –0.099 | –0.048 | –0.008 | 0.008 | –1.077 | 0.2835 | 119 |
| 10º N-10º S | +0.767 | 0.654 | 0.114 | 0.009 | 12.910 | <0.0001 | 119 |
| 10º-30º S | –0.472 | –0.273 | –0.048 | 0.008 | –5.799 | <0.0001 | 119 |
| 30º-60º S | –0.424 | –0.262 | –0.046 | 0.009 | –5.063 | <0.0001 | 119 |


| (b) GTO-ECV and QBO | Correlation | Intercept (%) | Slope* | Error | t-value | p-value | N |
|---|---|---|---|---|---|---|---|
| 30º-60º N | –0.116 | –0.090 | –0.012 | 0.007 | –1.869 | 0.0628 | 259 |
| 10º-30º N | –0.142 | –0.100 | –0.014 | 0.006 | –2.293 | 0.0226 | 259 |
| 10º N-10º S | +0.779 | 0.705 | 0.109 | 0.005 | 19.949 | <0.0001 | 259 |
| 10º-30º S | –0.484 | –0.306 | –0.046 | 0.005 | –8.873 | <0.0001 | 259 |
| 30º-60º S | –0.417 | –0.312 | –0.048 | 0.007 | –7.345 | <0.0001 | 259 |


| (b) SBUV v(8.6) and QBO | Correlation | Intercept (%) | Slope* | Error | t-value | p-value | N |
|---|---|---|---|---|---|---|---|
| 30º-60º N | –0.165 | –0.112 | –0.018 | 0.007 | –2.694 | 0.0075 | 262 |
| 10º-30º N | –0.177 | –0.114 | –0.018 | 0.006 | –2.901 | 0.0040 | 263 |
| 10º N-10º S | +0.748 | 0.648 | 0.104 | 0.006 | 18.223 | <0.0001 | 263 |
| 10º-30º S | –0.488 | –0.287 | –0.046 | 0.005 | –9.037 | <0.0001 | 263 |
| 30º-60º S | –0.458 | –0.328 | –0.051 | 0.006 | –8.333 | <0.0001 | 263 |


| (b) Ground-based and QBO | Correlation | Intercept (%) | Slope* | Error | t-value | p-value | N |
|---|---|---|---|---|---|---|---|
| 30º-60º N | –0.158 | –0.123 | –0.017 | 0.007 | –2.594 | 0.0100 | 264 |
| 10º-30º N | –0.142 | –0.083 | –0.016 | 0.007 | –2.317 | 0.0213 | 264 |
| 10º N-10º S | +0.695 | 0.553 | 0.095 | 0.006 | 15.327 | <0.0001 | 253 |
| 10º-30º S | –0.490 | –0.268 | –0.046 | 0.005 | –9.091 | <0.0001 | 264 |
| 30º-60º S | –0.431 | –0.322 | –0.048 | 0.006 | –7.734 | <0.0001 | 264 |

* The slope is in % per unit change of the explanatory variable. Error, t-value and p-value refer to slope.



**Table 4. Annual mean total ozone, amplitude of annual cycle, amplitude of QBO, amplitude of solar cycle and amplitude of ENSO in the period 1995-2016 from GOME-2A, GTO-ECV, the combined TOMS/OMI/OMPS satellite data and Oslo CTM3 model calculations over the South Pacific region (10°-20° S, 180°-220° E) and at station Samoa (14.25° S, 189.4° E) located within this region.**


| | South Pacific Ocean | | | | station Samoa | | | |
|---|---|---|---|---|---|---|---|---|
| | GOME-2A* | GTO-ECV | TOMS/OMI/OMPS | Oslo CTM3 | GOME-2A* | GTO-ECV | GROUND | SBUV (v8.6) |
| Annual mean | 255.3 DU | 254.7 DU | 253.0 DU | 259.5 DU | 252.7 DU | 252.2 DU | 249.2 DU | 251.9 DU |
| Amplitude of annual cycle | 7.4 DU (2.9%) | 7.7 DU (3.0%) | 7.3 DU (2.9%) | 5.2 DU (2.0%) | 7.1 DU (2.8%) | 6.7 DU (2.7%) | 6.7 DU (2.7%) | 7.3 DU (2.9%) |
| Amplitude of QBO | 2.7 DU (1.0%) | 2.2 DU (0.9%) | 2.4 DU (0.9%) | 2.3 DU (0.9%) | 3.0 DU (1.2%) | 2.2 DU (0.9%) | 2.7 DU (1.1%) | 2.0 DU (0.8%) |
| Amplitude of solar cycle | 2.1 DU (0.8%) | 4.1 DU (1.6%) | 4.6 DU (1.8%) | 1.8 DU (0.7%) | 2.0 DU (0.8%) | 4.5 DU (1.8%) | 1.6 DU (0.6%) | 4.5 DU (1.8%) |
| Amplitude of ENSO | 6.2 DU (2.4%) | 8.8 DU (3.5%) | 6.0 DU (2.4%) | 8.8 DU (3.4%) | 5.6 DU (2.2%) | 7.7 DU (3.0%) | 5.5 DU (2.2%) | 7.5 DU (3.0%) |

*period 2007-2016



**Table 5. Statistics of the comparisons between total ozone, tropopause pressures and SOI for a) South Pacific (10°-20° S, 180°-220° E), b) station Samoa (14.25° S, 189.4° E), c) South Asia (35°-45° N, 45°-125° E) and d) 7 stations in South Asia.**


| (a) South Pacific | Correlation with SOI | Intercept (%) | Slope* | Error | t-value | p-value | N |
|---|---|---|---|---|---|---|---|
| GOME-2A | +0.56 | –0.238 | 0.118 | 0.016 | 7.236 | <0.0001 | 119 |
| GTO-ECV | +0.66 | –0.069 | 0.145 | 0.010 | 14.014 | <0.0001 | 252 |
| TOMS/OMI/OMPS | +0.62 | –0.139 | 0.134 | 0.011 | 12.285 | <0.0001 | 241 |
| Oslo CTM3 | +0.55 | –0.064 | 0.144 | 0.014 | 10.501 | <0.0001 | 252 |
| Tropopause | +0.66 | –0.761 | 0.241 | 0.017 | 13.825 | <0.0001 | 252 |


| (b) Samoa | Correlation with SOI | Intercept (%) | Slope* | Error | t-value | p-value | N |
|---|---|---|---|---|---|---|---|
| GOME-2A | +0.47 | –0.217 | 0.108 | 0.018 | 5.823 | <0.0001 | 119 |
| GTO-ECV | +0.55 | –0.100 | 0.127 | 0.012 | 10.366 | <0.0001 | 252 |
| SBUV overpass | +0.59 | –0.114 | 0.127 | 0.011 | 11.398 | <0.0001 | 251 |
| GB (WOUDC) | +0.42 | –0.058 | 0.106 | 0.017 | 6.194 | <0.0001 | 178 |
| Tropopause | +0.65 | –0.799 | 0.223 | 0.017 | 13.405 | <0.0001 | 252 |


| (c) South Asia | Correlation with SOI | Intercept (%) | Slope* | Error | t-value | p-value | N |
|---|---|---|---|---|---|---|---|
| GOME-2A | –0.23 | 0.090 | –0.044 | 0.018 | –2.525 | 0.0129 | 119 |
| GTO-ECV | –0.30 | 0.073 | –0.074 | 0.015 | –5.047 | <0.0001 | 252 |
| TOMS/OMI/OMPS | –0.28 | –0.212 | –0.073 | 0.016 | –4.553 | <0.0001 | 241 |
| Oslo CTM3 | –0.18 | 0.140 | –0.040 | 0.014 | –2.877 | 0.0044 | 252 |
| Tropopause | –0.27 | –0.188 | –0.129 | 0.029 | –4.476 | <0.0001 | 252 |


| (d) South Asia (7 stations mean) | Correlation with SOI | Intercept (%) | Slope* | Error | t-value | p-value | N |
|---|---|---|---|---|---|---|---|
| GOME-2A | –0.23 | 0.090 | –0.043 | 0.017 | –2.518 | 0.0132 | 119 |
| GTO-ECV | –0.30 | 0.067 | –0.072 | 0.014 | –5.040 | <0.0001 | 252 |
| SBUV overpass | –0.27 | 0.086 | –0.066 | 0.015 | –4.464 | <0.0001 | 251 |
| GB (WOUDC) | –0.36 | 0.427 | –0.103 | 0.017 | –5.912 | <0.0001 | 240 |
| Tropopause | –0.28 | –0.122 | –0.160 | 0.035 | –4.597 | <0.0001 | 252 |

* The slope is in % per unit change of the explanatory variable. Error, t-value and p-value refer to slope.



**Table 6. Annual mean total ozone, amplitude of annual cycle, amplitude of QBO, amplitude of solar cycle and amplitude of NAO in the period 1995-2016 from GOME-2A, GTO-ECV, the combined TOMS/OMI/OMPS satellite data and Oslo CTM3 model calculations over the North Atlantic Ocean: (a) region 35°-50° N, 20°-50° W, and (b) region 15°-27° N, 30°-60° W.**


| | North Atlantic Ocean | | | | | | | |
| --- | --- | --- | --- | --- | --- | --- | --- | --- |
| | (a) 35°-50° N, 20°-50° W | | | | (b) 15°-27° N, 30°-60° W | | | |
| | GOME-2A* | GTO-ECV | TOMS/OMI/OMPS | Oslo CTM3 | GOME-2A* | GTO-ECV | TOMS/OMI/OMPS | Oslo CTM3 |
| Annual mean | 319.7 DU | 315.9 DU | 317.3 DU | 311.2 DU | 276.6 DU | 276.4 DU | 274.4 DU | 282.6 DU |
| Amplitude of annual cycle | 37.4 DU (11.7%) | 37.0 DU (11.7%) | 36.9 DU (11.6%) | 32.0 DU (10.3%) | 12.7 DU (4.6%) | 15.8 DU (5.7%) | 15.1 DU (5.5%) | 15.5 DU (5.5%) |
| Amplitude of QBO | 2.5 DU (0.8%) | 2.3 DU (0.7%) | 2.6 DU (0.8%) | 3.2 DU (1.0%) | 3.0 DU (1.1%) | 2.8 DU (1.0%) | 3.9 DU (1.4%) | 4.3 DU (1.5%) |
| Amplitude of solar cycle | 0.4 DU (0.1%) | 0.3 DU (0.1%) | 2.2 DU (0.7%) | 2.3 DU (0.7%) | 3.5 DU (1.3%) | 2.7 DU (1.0%) | 3.3 DU (1.2%) | 1.0 DU (0.3%) |
| Amplitude of NAO (winter) | 18.3 DU (5.7%) | 16.5 DU (5.2%) | 18.4 DU (5.8%) | 18.3 DU (5.9%) | 4.2 DU (1.5%) | 7.2 DU (2.6%) | 5.0 DU (1.8%) | 8.0 DU (2.8%) |

*period 2007-2016



**Table 7. Annual mean total ozone, amplitude of annual cycle, amplitude of QBO, amplitude of solar cycle and amplitude of NAO in the period 1995-**
**2016 from GOME-2A, GTO-ECV satellite data, ground-based observations and SBUV (v8.6) satellite overpass data over: (a) Canada and USA (11**
**stations mean), and (b) Europe (41 stations mean).**

|  | (a) Canada and USA | | | | (b) Europe | | | |
|---|---|---|---|---|---|---|---|---|
|  | 30º-50º N, 60º-110º W (11 stations mean) | | | | 35º-55º N, 10º W-40º E (41 stations mean) | | | |
|  | GOME-2A* | GTO-ECV | GROUND | SBUV (v8.6) | GOME-2A* | GTO-ECV | GROUND | SBUV (v8.6) |
| Annual mean | 324.2 DU | 320.6 DU | 322.5 DU | 320.9 DU | 329.9 DU | 325.7 DU | 326.9 DU | 326.8 DU |
| Amplitude of annual cycle | 38.1 DU (11.7%) | 34.1 DU (10.6%) | 33.2 DU (10.3%) | 34.0 DU (10.6%) | 39.3 (11.9%) | 40.5 DU (12.4%) | 39.2 DU (12.0%) | 40.7 DU (12.4%) |
| Amplitude of QBO | 2.1 DU (0.6%) | 2.5 DU (0.8%) | 3.5 DU (1.1%) | 2.6 DU (0.8%) | 2.7 DU (0.8%) | 1.9 DU (0.6%) | 2.8 DU (0.8%) | 2.2 DU (0.7%) |
| Amplitude of solar cycle | 0.3 DU (0.1%) | 0.5 DU (0.2%) | 1.4 DU (0.4%) | 0.5 DU (0.2%) | 2.1 DU (0.6%) | 0.8 DU (0.2%) | 1.0 DU (0.3%) | 0.3 DU (0.1%) |
| Amplitude of NAO (winter) | 9.8 DU (3.0%) | 6.9 DU (2.2%) | 8.7 DU (2.7%) | 9.3 DU (2.9%) | 9.8 DU (3.0%) | 8.9 DU (2.7%) | 11.8 DU (3.6%) | 9.9 DU (3.0%) |

*period 2007-2016



**Table 8. Statistics of the comparisons between total ozone, tropopause pressures and NAO index in winter (DJF mean) for a) the northern part of North Atlantic (35$^o$-50$^o$ N, 20$^o$-50$^o$ W), b) its southern part (15$^o$-27$^o$ N, 30$^o$-60$^o$ W), c) 11 stations in Canada and USA, and d) 41 stations in Europe.**


| (a) Northern part of North Atlantic | Correlation with NAO in winter | Intercept (%) | Slope* | Error | t-value | p-value | N |
|---|---|---|---|---|---|---|---|
| GOME-2A | −0.85 | 0.035 | −2.474 | 0.568 | −4.355 | 0.0033 | 9 |
| GTO-ECV | −0.74 | 0.412 | −2.188 | 0.453 | −4.827 | 0.0001 | 21 |
| TOMS/OMI/OMPS | −0.74 | 0.734 | −2.386 | 0.538 | −4.436 | 0.0004 | 18 |
| Oslo CTM3 | −0.75 | 0.639 | −2.457 | 0.498 | −4.937 | <0.0001 | 21 |
| Tropopause | −0.83 | 0.665 | −3.112 | 0.480 | −6.478 | <0.0001 | 21 |


| (b) Southern part of North Atlantic | Correlation with NAO in winter | Intercept (%) | Slope* | Error | t-value | p-value | N |
|---|---|---|---|---|---|---|---|
| GOME-2A | +0.54 | −0.132 | 0.661 | 0.386 | 1.712 | 0.1306 | 9 |
| GTO-ECV | +0.60 | −0.202 | 1.097 | 0.333 | 3.291 | 0.0038 | 21 |
| TOMS/OMI/OMPS | +0.58 | −0.334 | 1.138 | 0.402 | 2.832 | 0.0120 | 18 |
| Oslo CTM3 | +0.65 | −0.077 | 1.188 | 0.316 | 3.761 | 0.0013 | 21 |
| Tropopause | +0.59 | −0.702 | 1.547 | 0.482 | 3.207 | 0.0046 | 21 |


| (a) CA/USA (11 stations mean) | Correlation with NAO in winter | Intercept (%) | Slope* | Error | t-value | p-value | N |
|---|---|---|---|---|---|---|---|
| GOME-2A | −0.71 | −0.042 | −1.305 | 0.493 | −2.647 | 0.0331 | 9 |
| GTO-ECV | −0.40 | 0.308 | −0.904 | 0.479 | −1.886 | 0.0746 | 21 |
| SBUV overpass | −0.50 | 0.318 | −1.209 | 0.476 | −2.541 | 0.0199 | 21 |
| GB (WOUDC) | −0.46 | 0.268 | −1.046 | 0.477 | −2.190 | 0.0419 | 20 |
| Tropopause | −0.41 | 0.268 | −0.739 | 0.377 | −1.959 | 0.0650 | 21 |


| (b) Europe (41 stations mean) | Correlation with NAO in winter | Intercept (%) | Slope* | Error | t-value | p-value | N |
|---|---|---|---|---|---|---|---|
| GOME-2A | −0.46 | 0.089 | −1.282 | 0.897 | −1.428 | 0.1963 | 9 |
| GTO-ECV | −0.42 | 0.315 | −1.141 | 0.573 | −1.992 | 0.0609 | 21 |
| SBUV overpass | −0.47 | 0.389 | −1.264 | 0.543 | −2.329 | 0.0311 | 21 |
| GB (WOUDC) | −0.48 | 0.625 | −1.327 | 0.560 | −2.368 | 0.0287 | 21 |
| Tropopause | −0.40 | 0.048 | −0.989 | 0.523 | −1.891 | 0.0739 | 21 |

* The slope is in % per unit change of the explanatory variable. Error, t-value and p-value refer to slope,



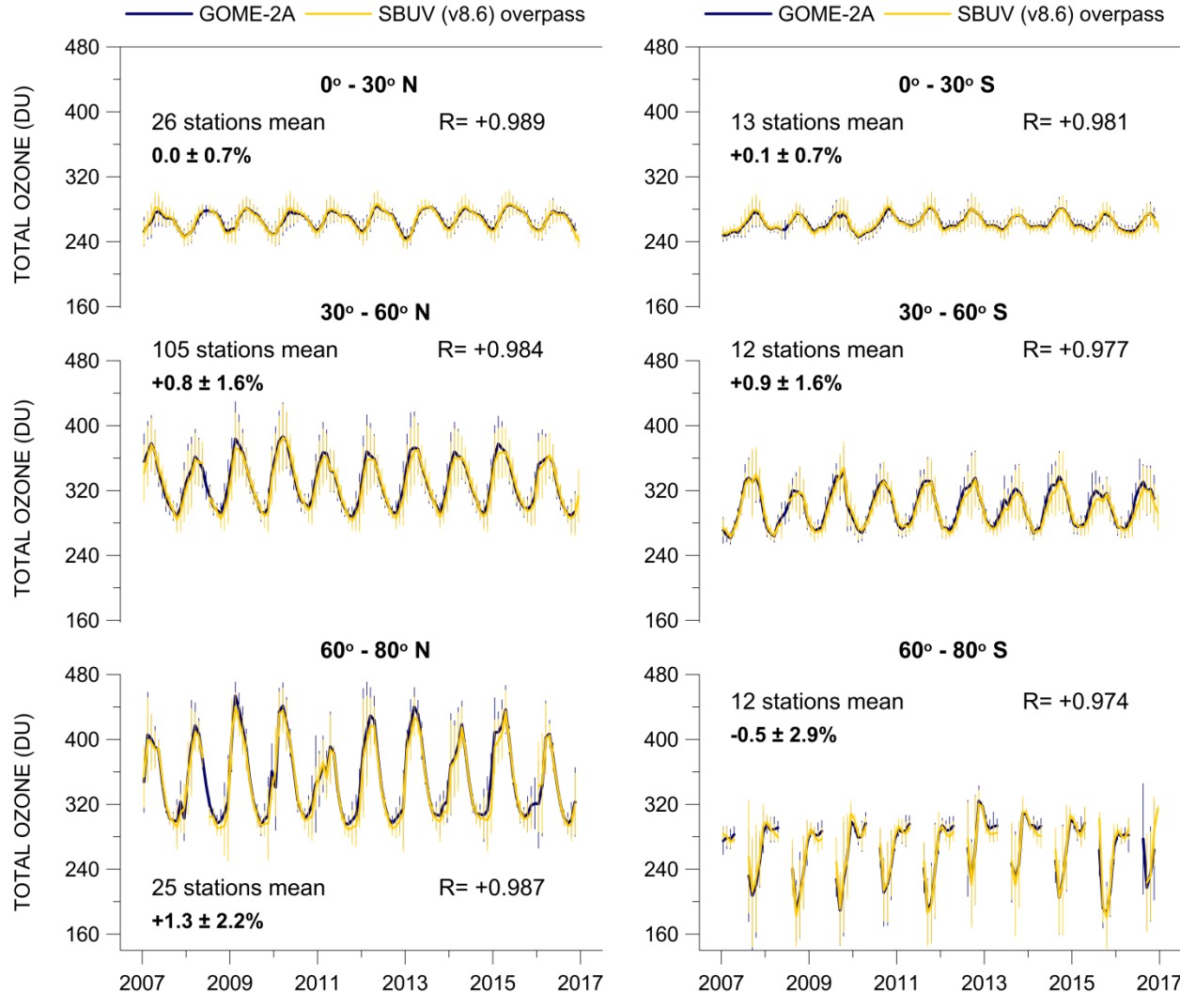



Figure 1. Monthly mean total ozone from GOME-2A as compared with monthly mean total ozone from SBUV (v8.6) satellite overpass data for the period 2007-2016 over the Northern and the Southern Hemisphere based on stations mean data. *R* is the correlation coefficient between the two lines. Error bars show the standard deviation of each monthly mean. Mean differences ± σ are given as [GOME-2A – SBUV] / SBUV (%).




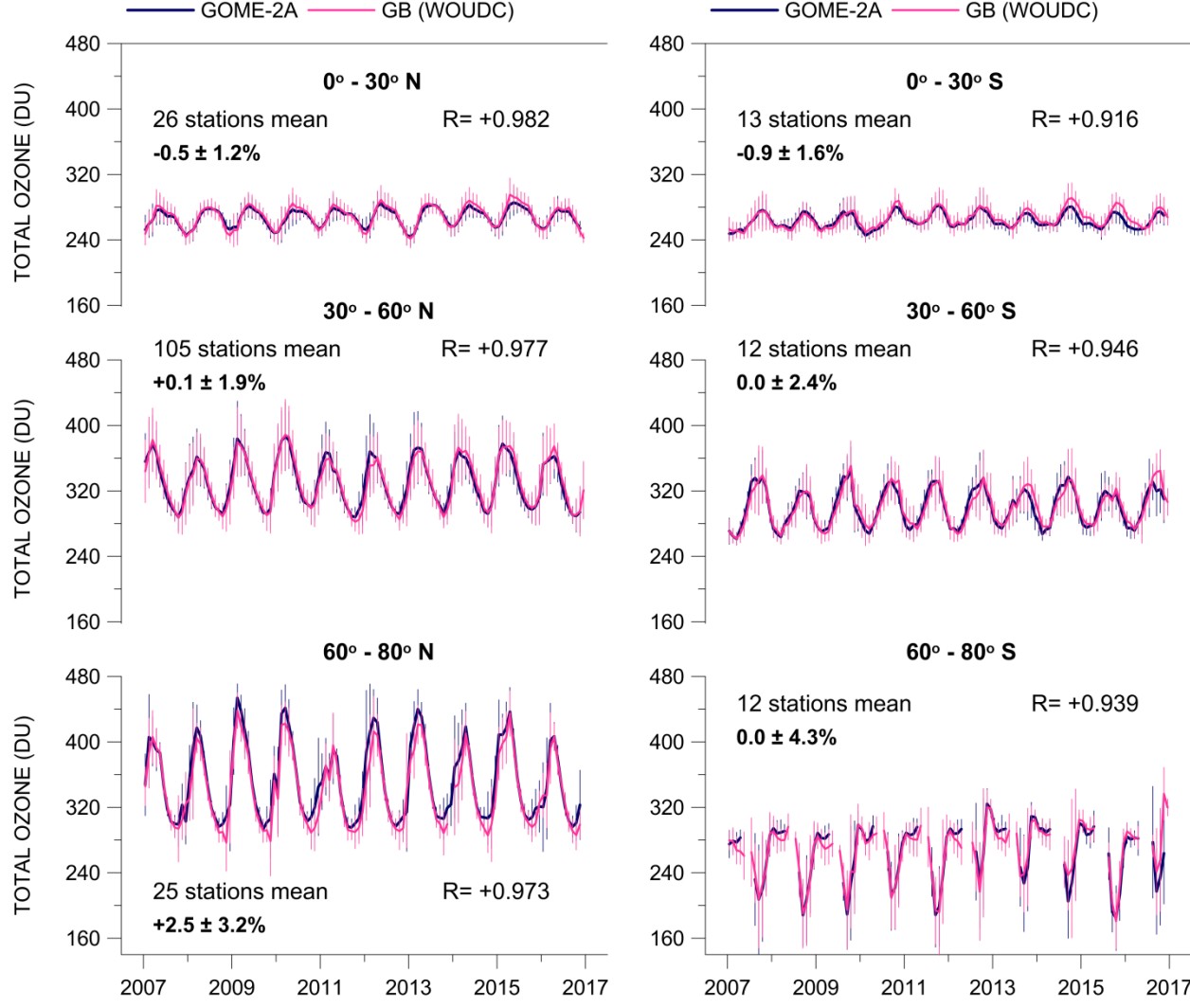



**Figure 2. Same as in Figure 1 but for GOME-2A and GB observations. *R* is the correlation coefficient between the two lines. Error bars show the standard deviation of each monthly mean. Mean differences ± σ are given as [GOME-2A – GROUND] / GROUND (%).**



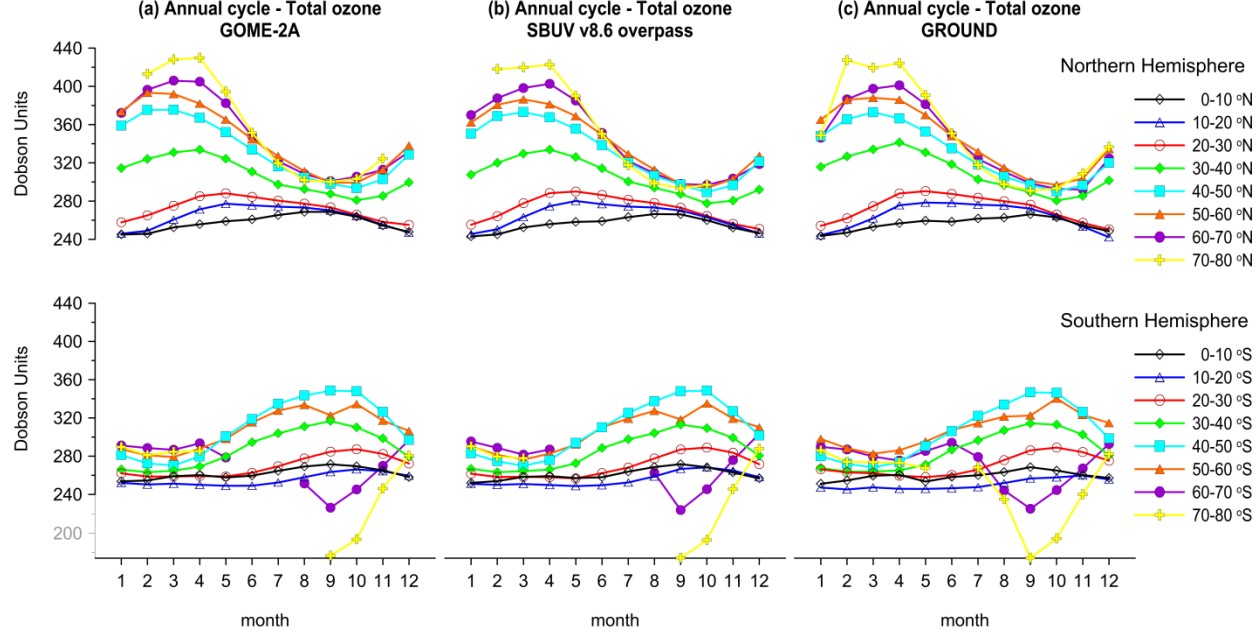

**Figure 3. Comparison of the annual cycle of total ozone from GOME-2A with that from SBUV (v8.6) satellite overpass data and GB observations in the period 2007-2016 based on stations data averaged per 10 degree latitude zones. The annual cycle is distorted above 60 deg. S due to the Antarctic ozone hole.**


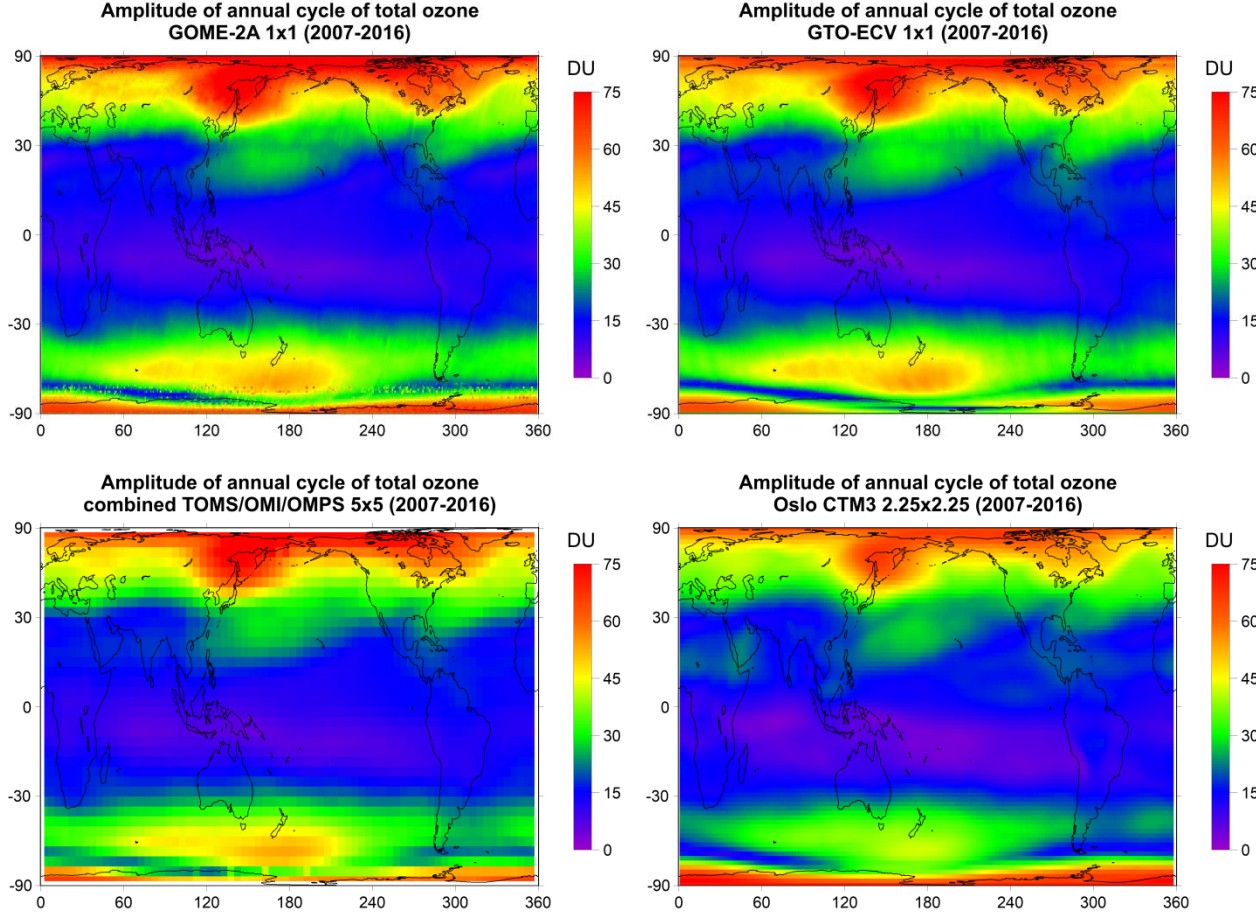


**Figure 4. Comparison of the amplitude [i.e., (max-min)/2] of the annual cycle of total ozone from GOME-2A (upper left) with the amplitude of the annual cycle of total ozone from GTO-ECV (upper right), the combined TOMS/OMI/OMPS satellite data (lower left) and Oslo CTM3 model simulations (lower right).**




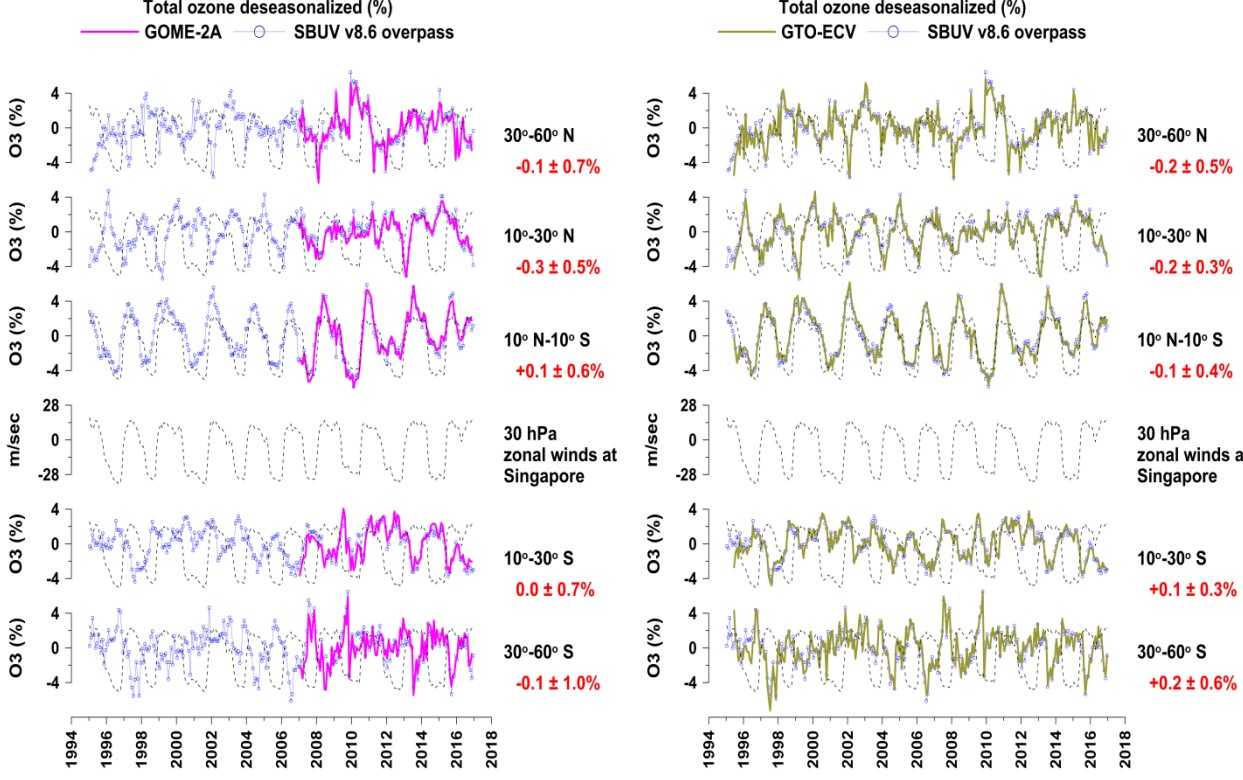


**Figure 5. (Left panel) Time series of deseasonalised total ozone from GOME-2A and SBUV (v8.6) satellite overpasses over different latitude zones along with the equatorial zonal winds at 30 hPa as an index of the QBO; (Right panel) same as in left panel but for GTO-ECV and SBUV. Values with red colour refer to the mean differences ± σ (in %) between GOME-2A and SBUV deseasonalised data averaged over various WOUDC stations (150 stations in the northern mid-latitudes (30°-60° N), 21 stations in the northern subtropics (10°-30° N), 8 stations in the tropics (10° S-10° N), 10 stations in southern subtropics (10°-30° S) and 12 stations in the southern mid-latitudes (30°-60° S)). The QBO proxy is superimposed on the ozone anomalies.**



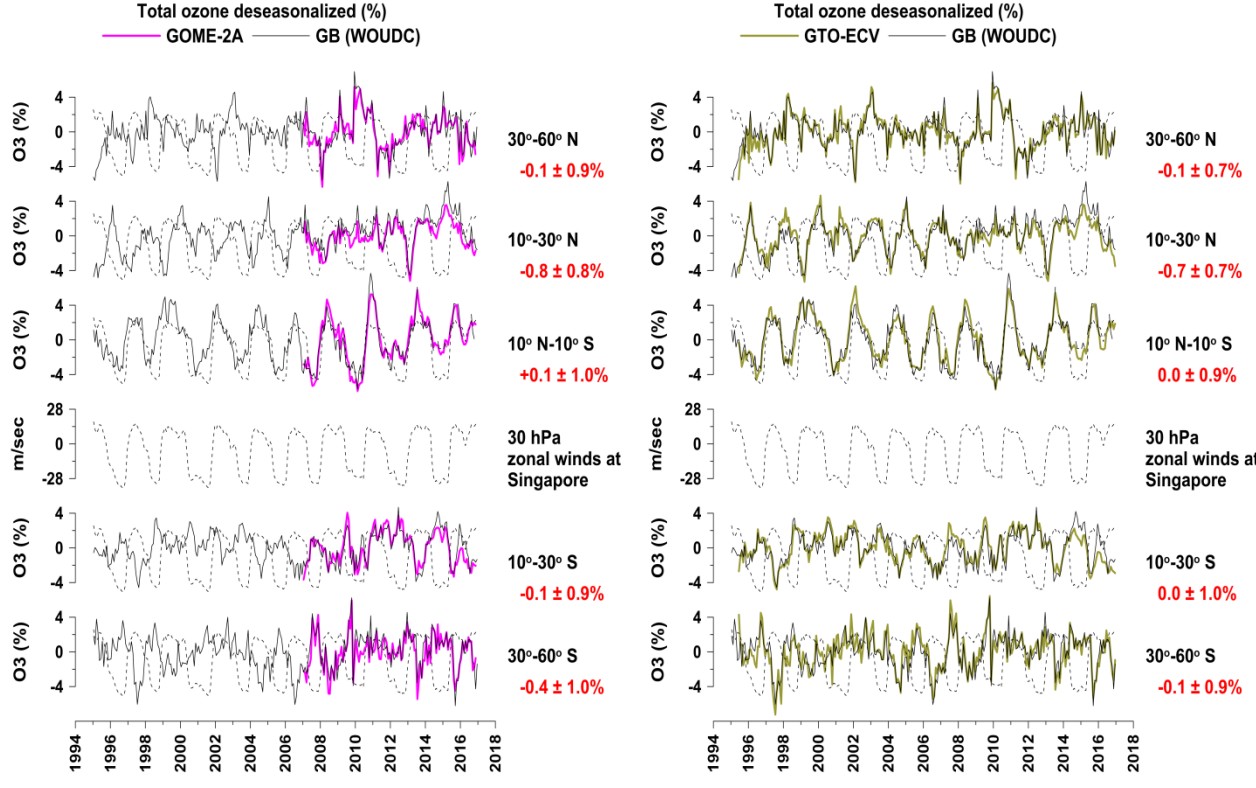


**Figure 6. Same as in Figure 5 but for GOME-2A and GB observations (left panel), and for GTO-ECV and GB observations (right panel). The QBO proxy is superimposed on the ozone anomalies.**



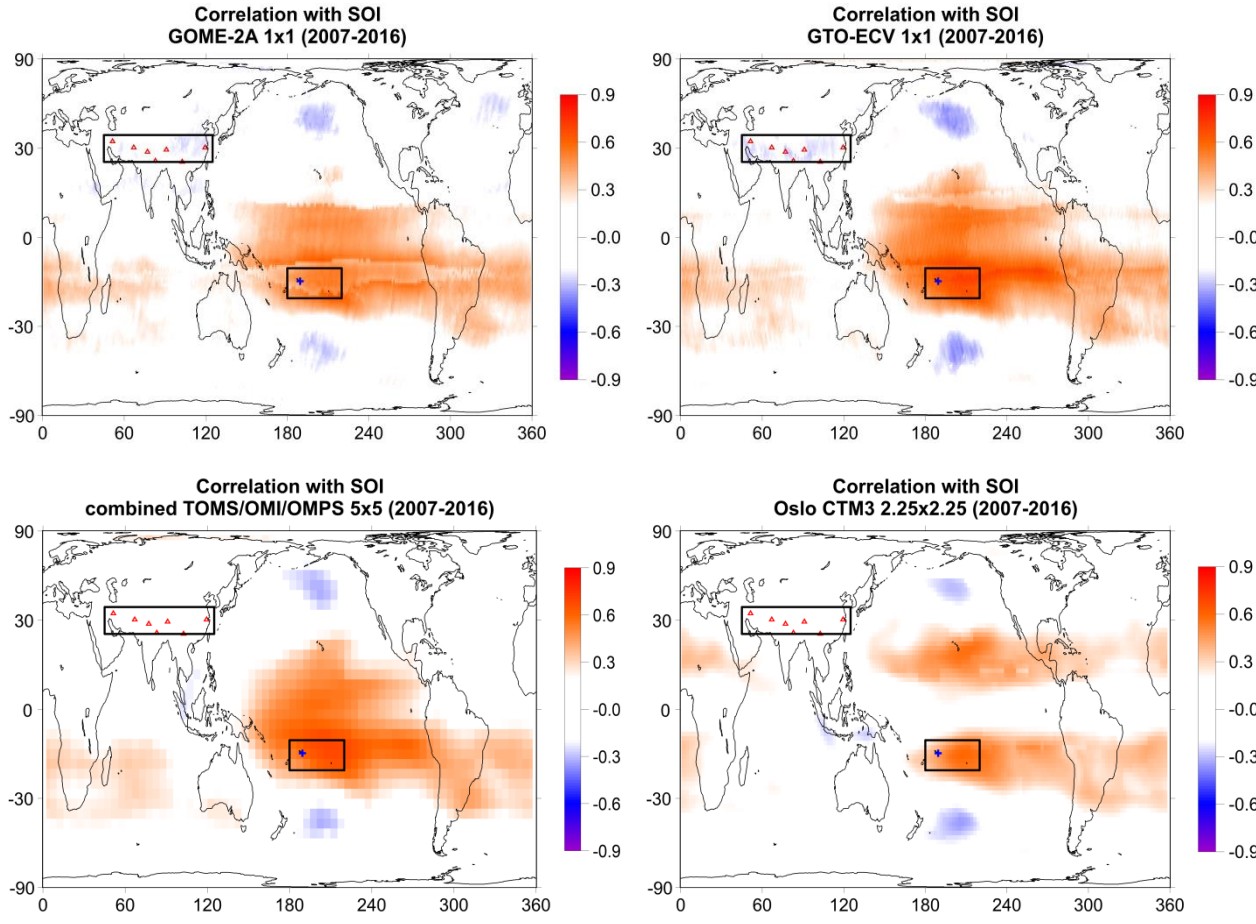


**Figure 7. Map of correlation coefficients between total ozone and SOI for GOME-2A (upper left), GTO-ECV (upper right), TOMS/OMI/OMPS satellite data (lower left) and Oslo CTM3 model simulations (lower right). Rectangles correspond to the South Pacific region (10-20 ºS, 180-220 ºE) and South Asia region (35-45 ºN, 45-125 ºE), blue cross to the station Samoa (14.25 ºS, 189.4 ºE) and red triangles to stations in South Asia, in which total ozone has been studied as for the impact of ENSO after removing variability related to the annual cycle, QBO and solar cycle. Positive correlations are shown by red colours while negative correlations by blue colours. Only correlation coefficients above/below ±0.2 are shown.**

920
921

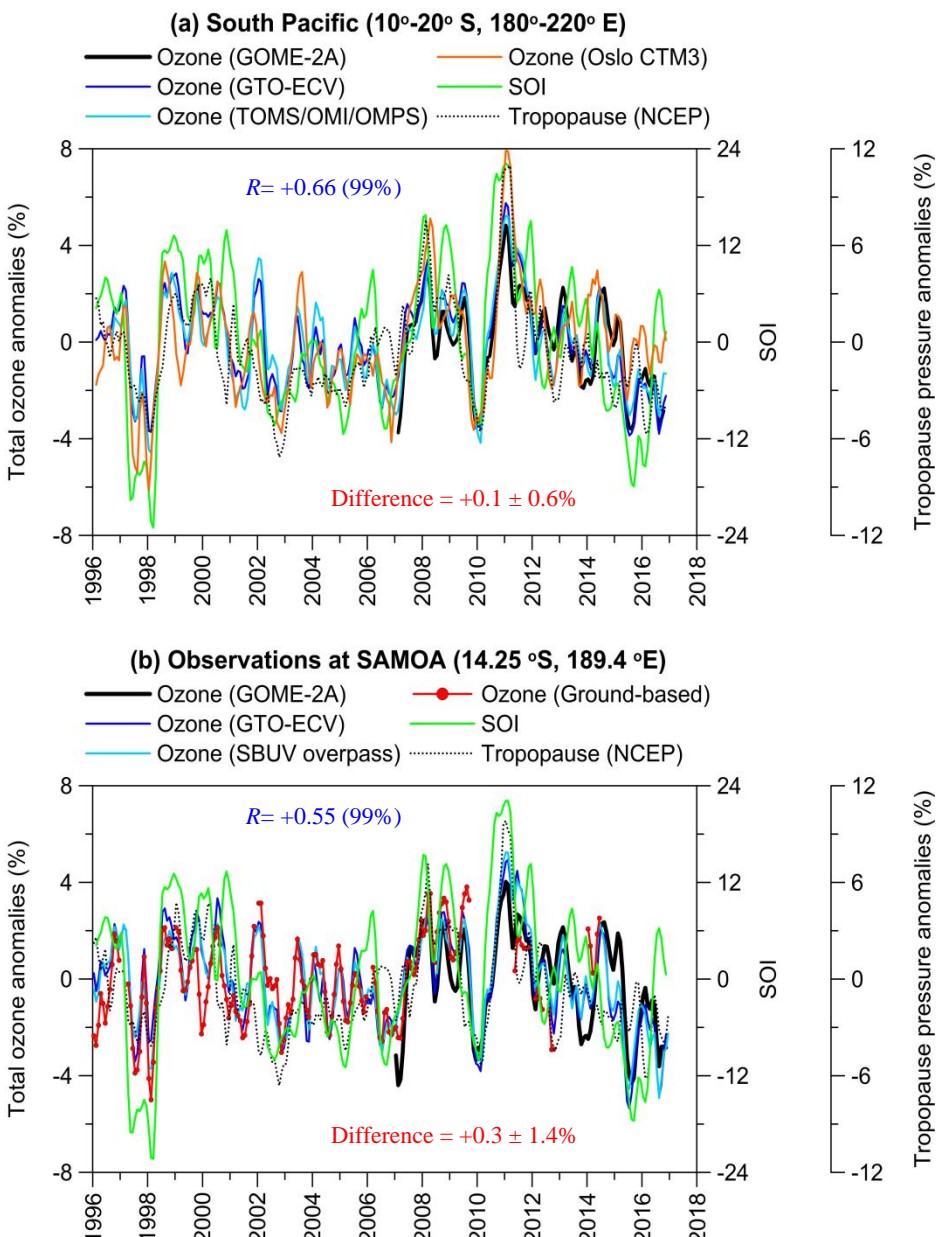

922

**Figure 8. (a) Example of regional time series of total ozone (%) over the South Pacific region (10°-20° S, 180°-220° E) along with SOI. The dotted line shows the respective tropopause pressure variability from NCEP. *R* is the correlation coefficient between GTO-ECV total ozone and SOI (statistical significance of *R* is given in parentheses). The difference refers to the mean difference ± σ (in %) between GTO-ECV and the combined TOMS/OMI/OMPS satellite data. (b) Same as in (a) but for SBUV overpass and GB data at the station Samoa. The difference refers to the mean difference ± σ (in %) between GTO-ECV and GB data.**

929
930

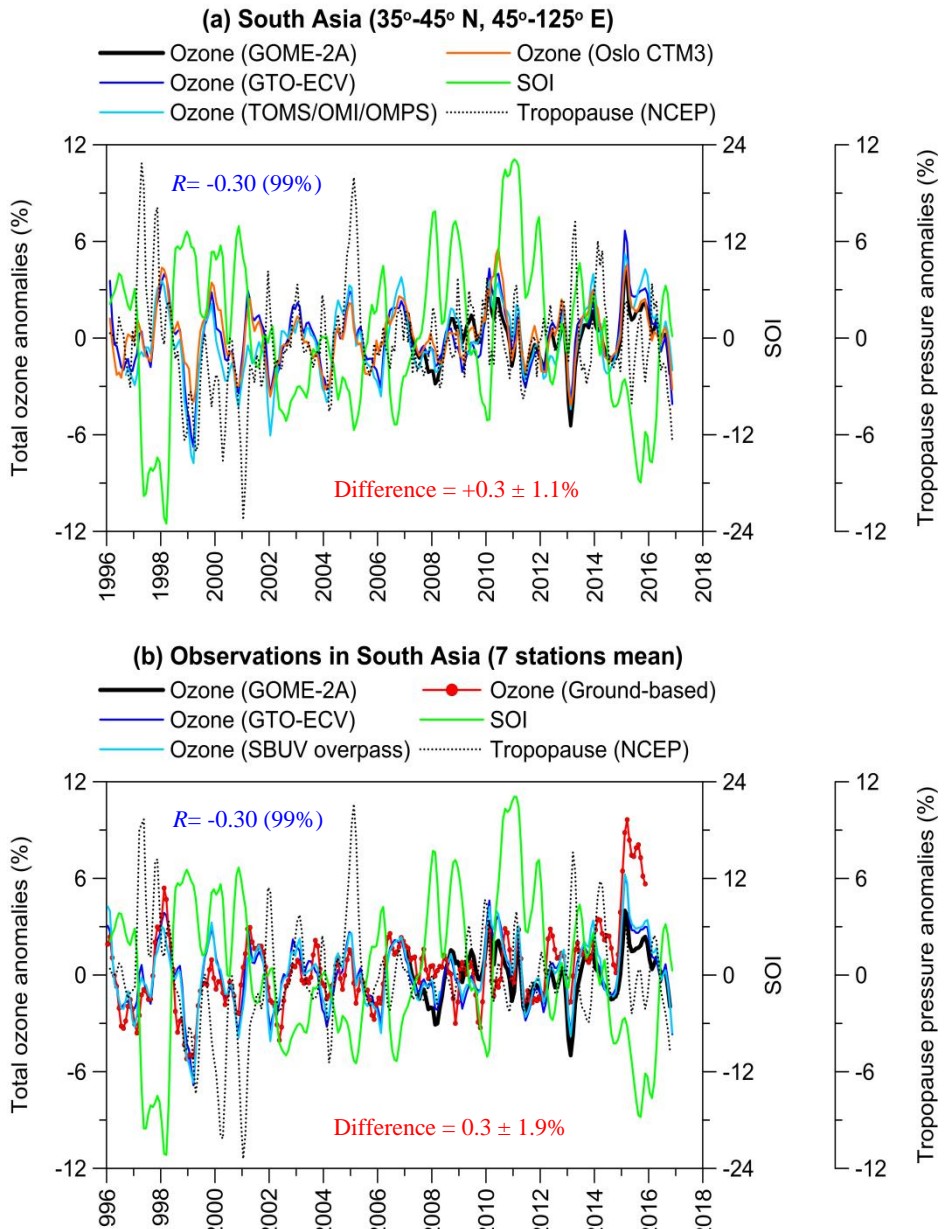

931

**Figure 9. (a) Example of regional time series of total ozone (%) over South Asia (35°-45° N, 45°-125° E) along with SOI. The dotted line shows the respective tropopause pressure variability from NCEP. *R* is the correlation coefficient between GTO-ECV total ozone and SOI (statistical significance of *R* is given in parentheses). The difference refers to the mean difference ± σ (in %) between GTO-ECV and the combined TOMS/OMI/OMPS satellite data. (b) Same as in (a) but with SBUV overpass and GB data averaged at 7 stations in South Asia. The difference refers to the mean difference ± σ (in %) between GTO-ECV and GB data.**



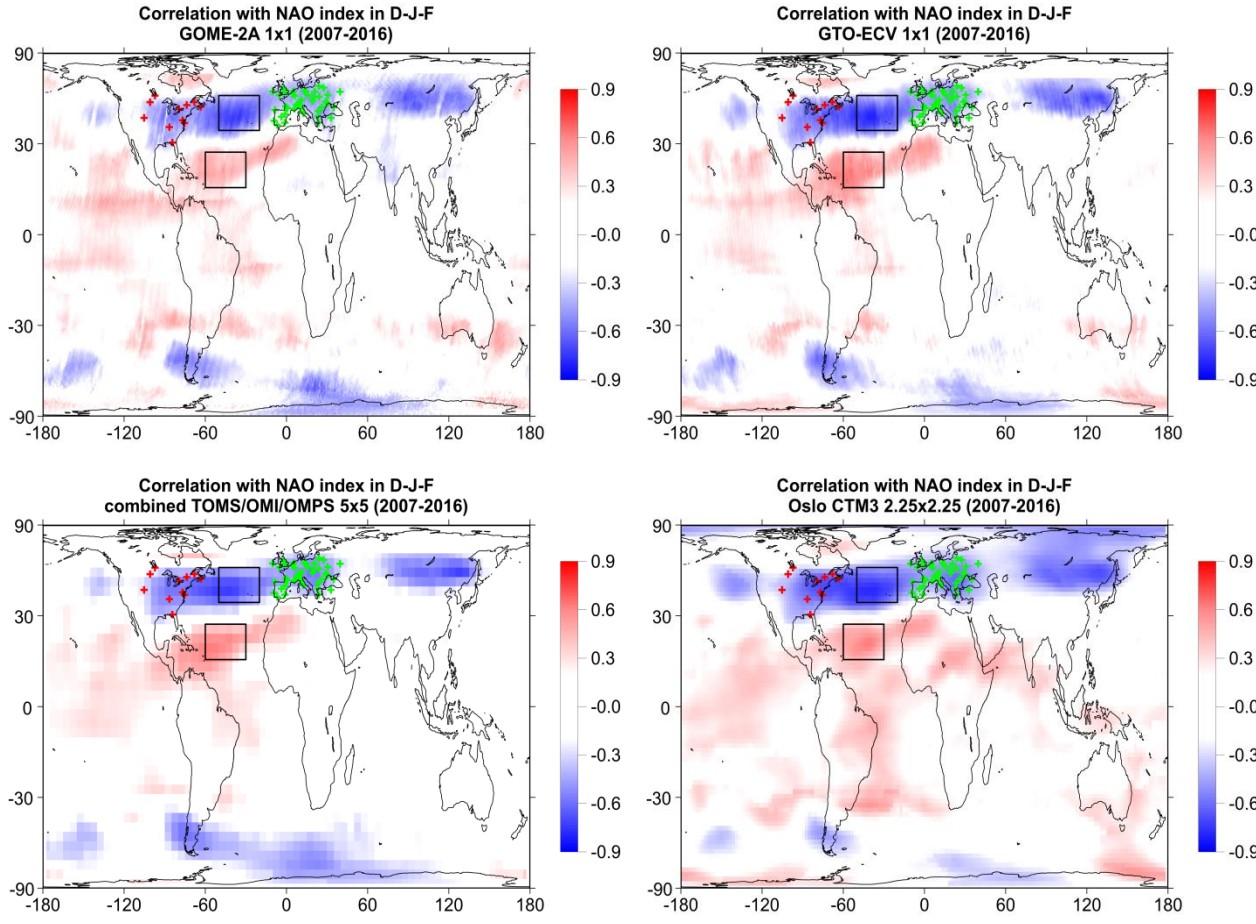


**Figure 10. Map of correlation coefficients between total ozone and the NAO index during winter (December, January, February; D-J-F) for GOME-2A (upper left), GTO-ECV (upper right), TOMS/OMI/OMPS satellite data (lower left) and Oslo CTM3 model simulations (lower right). Rectangles correspond to regions in the North Atlantic (35°-50° N, 20°-50° W; 15°-27° N, 30°-60° W), and red and green crosses to stations in Canada/USA and Europe, in which total ozone has been studied as for the impact of NAO after removing variability related to the annual cycle , QBO, solar cycle and ENSO. Positive correlations are shown by red colours while negative correlations by blue colours. Only correlation coefficients above/below ±0.2 are shown.**



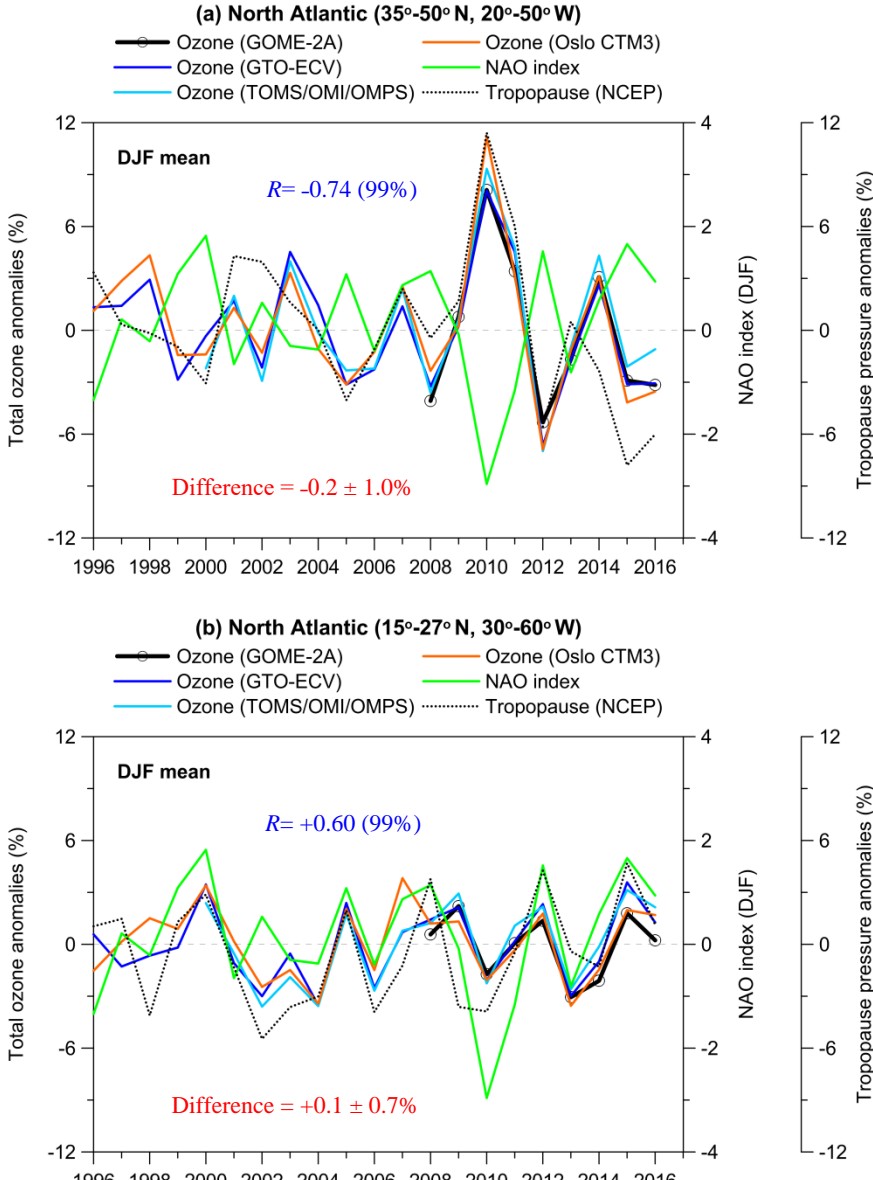


**Figure 11. Example of regional time series of total ozone (%) over the North Atlantic regions (a) 35°-50° N, 20°-50° W and (b) 15°-27° N, 30°-60° W in winter (DJF mean) along with the NAO index. The dotted line shows the respective tropopause pressure variability from NCEP reanalysis. *R* is the correlation coefficient between GTO-ECV total ozone and the NAO index. The differences refer to the mean differences ± σ (in %) between GTO-ECV and the combined TOMS/OMI/OMPS satellite data.**



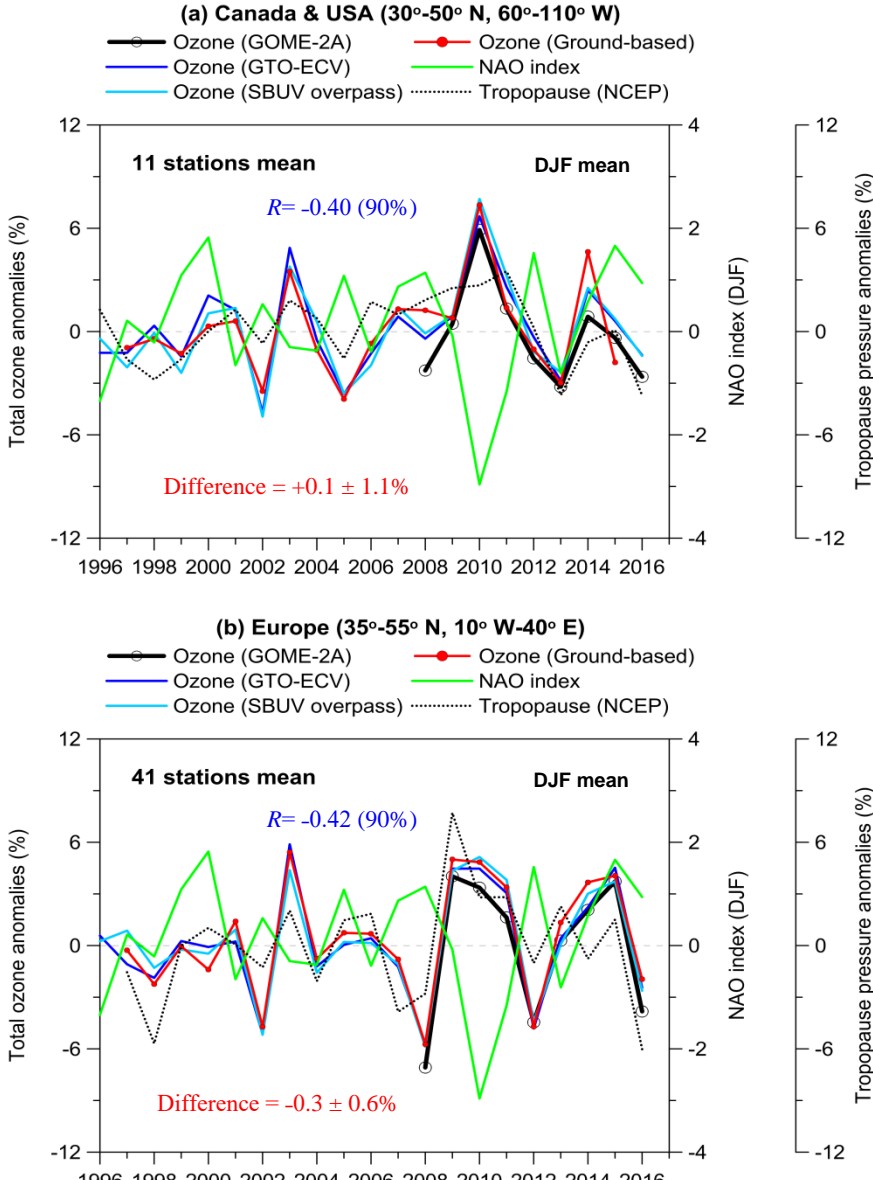


**Figure 12. Comparison with GB observations over: (a) Canada and USA and (b) Europe in winter (DJF mean).** *R* **is the correlation coefficient between GTO-ECV total ozone and the NAO index. The differences refer to the mean differences ± σ (in %) between GTO-ECV and GB data.**




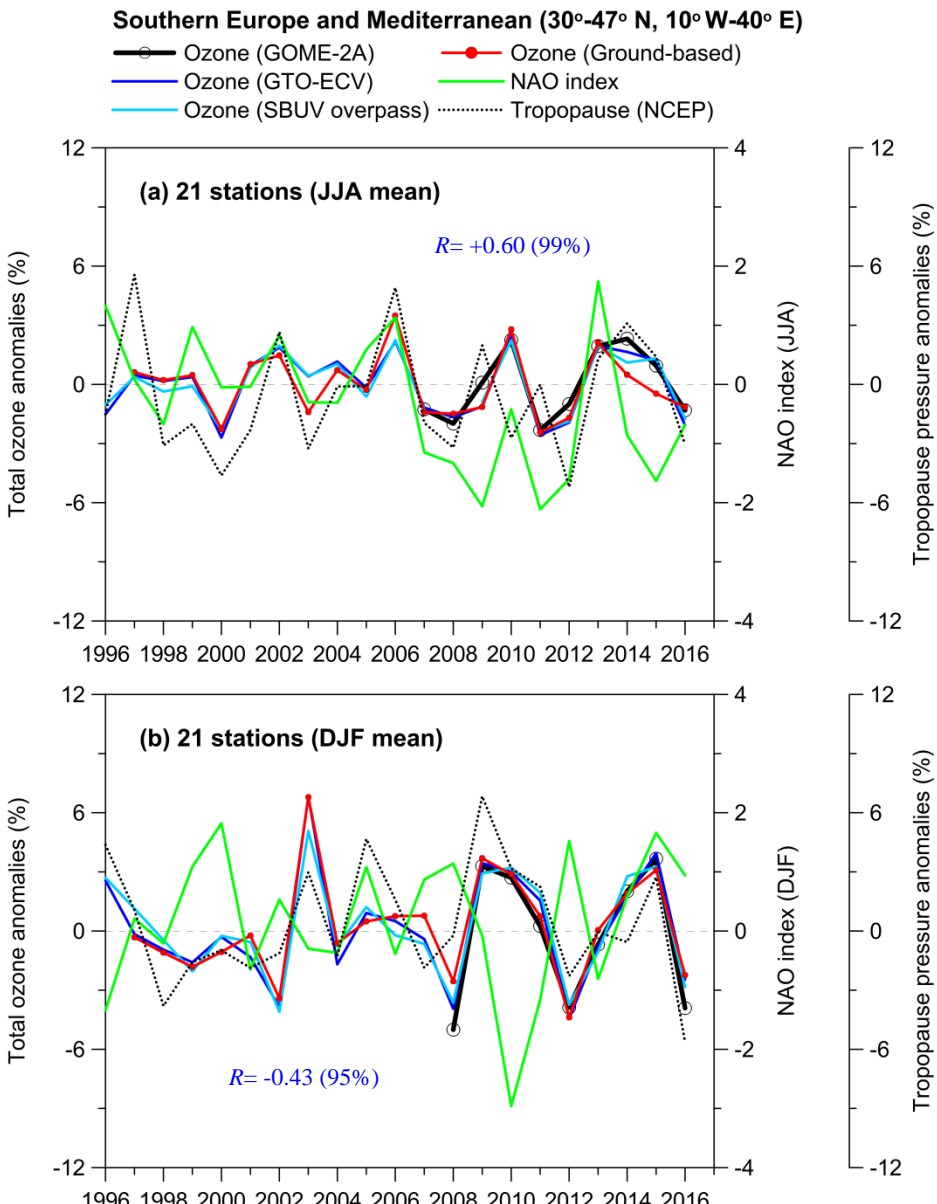



**Figure 13. Relation between total ozone and the NAO index in summer (JJA mean) and winter (DJF mean)**
**for 21 stations in southern Europe. The correlation coefficients refer to NAO index and GB total ozone after**
**removing variability related to the seasonal cycle, QBO, solar cycle and ENSO.**

