# Peer review of "The use of QBO, ENSO and NAO perturbations in the evaluation of GOME-2/MetopA total ozone measurements"

_Atmospheric Measurement Techniques, 2018_

## Referee Comment (RC1) · Anonymous Referee #1 · 28 Sep 2018

In this paper some interesting analyses are presented about the QBO, ENSO and NAO signal in various long-term ozone data sets. However, I have some problems to find the main aim of the paper. In the abstract it is mentioned that validation is performed for GOME2-A, yet no direct comparison with ground observations has been made. The correlations have been derived for QBO, ENSO and NAO signals, which although interesting as it is, I would not call validation. The term "evaluation" mentioned in the title is a better description. In the title, on the other hand, only GOME-2A is mentioned, while the authors are evaluating SBUV and GTO-ECV in exactly the same way. I suggest to change the title to "The use of QBO, ENSO and NAO perturbations in the evaluation of long-term total ozone satellite measurements." and to use 'evaluation'

instead of 'validation' throughout the text.

Throughout the paper correlations are calculated for the comparisons, which I think is very limited. I suggest that the authors provide more information on these comparisons by calculating the regression coefficients.

Detailed comments:

Line 23: validating => evaluating

Line 29: Here the GTO-ECV data set is mentioned for the first time. I don't think most readers will have a clear idea what "GOME-type Total Ozone Essential Climate Variable" mean. A short description to describe this data set would be helpful at this point.

Line 51: Cause&effect are reversed in this sentence. Ozone is considered a greenhouse gas because it warms the Earth's surface not the other way around. In addition, it might be good to mention that not only tropospheric ozone but also stratospheric ozone is a greenhouse gas.

Line 56: "launched in 2018." => "launched end of 2018"

Line 73: Except for the abstract, this is the first time that the SBUV and GTO-ECV data sets are mentioned, therefore, I suggest to add references for both data sets in the text.

Line 89-91: It might be better to refer to more recent papers about the recovery of the ozone layer, for example de Laat et al., Onset of Stratospheric Ozone Recovery in the Antarctic Ozone Hole in Assimilated Daily Total Ozone Columns, JGR, 2017, https://doi.org/10.1002/2016JD025723

Line 150-151: When mentioning the various long-term data sets of ozone, also the Multi-Sensor Reanalysis of ozone comes to mind. This data set has also been analysed for QBO, ENSO, NAO and other perturbations in Knibbe et al., ACP, 2014 and therefore is worthwhile to include here and in the discussion at the end of section 3.3.

Line 223: I prefer to see more than only correlation coefficients. The regression parameters could be given here and in the remainder of the analyses.

Line 239-240: This sentence seems to saying that the origin of the blue zone (i.e. small amplitude) is attributed to the small amplitude in these parts. Please, give the real origin if this is known.

Line 259-265: This analysis was already discussed in section 3.1. Only this time the monthly mean has been subtracted which does not really change the validation. I suggest to remove this or add it in section 3.1

Line 269: A clear phase shift in Figure 5 is mentioned for higher latitudes. Actually for SBUV I see an anti-correlation with the phase for latitudes between -10 and 10, and for GOME2 I see neither phase shift or an anticorrelation. So I would not call this a clear phase shift. A discussion about the clear differences in result of SBUV (pre 2008) and GOME-2 should be added here as well.

Line 291-292: The correlations are not removed but the relation between ozone and QBO has been removed. Please, reformulate.

Line 295: If you are using this equation, it would be very interesting to mention also the fitted a0 and a1 instead or in addition to the found correlations.

Section 3.3, Figure 8 and 9: the GOME2 values in the last 4 year of the Figures 8 and 9 show a much worse comparison than the other years in the time series. Is there any explanation for this? I miss this in the discussion of the results here.

Line 367: A discussion of a comparison with the work of Knibbe et al., ACP, 2014 would be useful at this point.

Line 370: Here the effects of QBO are removed, but what about the ENSO perturbations? Are these also removed before continuing studying the NAO effects. The two effects have to be separated.

[Figure]

Line 293-393: Same as previous remark.

Line 469: This is not a real validation because a lot is still unknown about the quantification of the QBO, ENSO and NAO, therefore it is qualitative evaluation not a quantitative validation resulting in uncertainty estimates.

Figure 1: It is very difficult to distinguish the GOME2-A line and the SBUV-line. The legend doesn't seem to be correct either?

---

## Referee Comment (RC2) · Anonymous Referee #2 · 4 Oct 2018

General comments

The study of Eleftheratos et al. evaluates the performance of different satellite datasets, giving particular emphasis on the GOME2/MetOpA, of the total ozone column (TOC) based on their ability to depict natural fluctuations, such as QBO, ENSO and NAO. In addition it provides an updated validation of the satellite datasets by comparison with ground based TOC measurements, using all the available networks and instruments at WOUDC (i.e. Brewer, Dobson, Filter radiometers etc). The paper is well written overall, making extensive use of the available literature and the results are visualized in an appropriate way, although they could have improvements (see specific comments

below). It is an interesting addition in the studies for the continuous evaluation of the good performance of satellite platforms, fits well on the scope of the journal and thus I suggest to be accepted after some minor changes.

Specific comments

Figures 1 and 2: Maybe you could select a different combination of colors since now it is hard the differences between the different datasets to be distinguished. Alternatively you could plot the monthly differences, instead of the actual total ozone column values.

Page 6 lines 214-215: It is mentioned that the highest differences are found over the southern high latitudes, however from Figures 1 and 2 it is depicted that these are presented over the Northern high latitudes (60 – 80 N) and the highest variability (standard deviation of the mean difference) is observed over the latitude belt (60 – 80 S). In addition, these differences (especially at the high latitudes) can be affected by the fact that you have not used the same days for the construction of the monthly mean values for the different datasets.

Page 7 lines 220-226: Which statistical test did you use to check the statistical significance?

Page 8 lines 269 – 271: I don't think that you see the amplitude of QBO effect on your total ozone column. The times series are just deseasonalized, but still contain the effect of other signals such as the 11 year solar cycle, ENSO etc and thus not all the variation can be attributed to QBO.

Figures 5 and 6: You could possible superimpose the QBO proxy on the ozone anomalies.

Section 3.3: You removed the effect of the annual cycle and QBO, before you correlate your ozone time series with ENSO but the effect of solar cycle could also affect your results.

Page 9 lines 306-307: Which statistical test did you use for checking the statistical
significance?

Section 3.4: Here you discuss the correlations between total ozone column and the NAO during winter months, evaluating the known anti-correlation between those two factors. Maybe it would be of interest to look also the correlations during summer, following the study of Osso et al. who reported a reversal in the correlation pattern between NAO and TOC from winter to summer for southern Europe.

Ossó A, Sola Y, Bech J, Lorente J (2011) Evidence for the influence of the North Atlantic Oscillation on the total ozone column at northern low latitudes and midlatitudes during winter and summer seasons. J Geophys Res Atmos 116:D24122. doi: 10.1029/2011JD016539

Typos:

Page 5, line 146: 5o -> 5º

Page 5, line 149: all offsets where -> all offsets were

Page 5, line 179: we made use of the monthly -> we used the monthly

Page 6, line 181: we made use of the monthly -> we used the monthly

Page 6, lines 187 – 190: "Use was made of the principal . . ." doesn't sound very nice maybe you could change to: "The principal component (PC)-based NAO index (DJF) provided by the . . . (last access: 15 June 2018) was used (or analyzed).

Page 6, line 190: After dynamical variability add ","

Page 6, line 192: The impact of tropopause variability on -> The impact of the tropopause height variations on

---

## Author Comment (AC1) · 21 Dec 2018

**Answers to Comments of Reviewer 1**

We would like to thank the reviewer for the fruitful comments and suggestions which helped improving the manuscript.

**General comment 1**

In this paper some interesting analyses are presented about the QBO, ENSO and NAO signal in various long-term ozone data sets. However, I have some problems to find the main aim of the paper. In the abstract it is mentioned that validation is performed for GOME2-A, yet no direct comparison with ground observations has been made. The correlations have been derived for QBO, ENSO and NAO signals, which although interesting as it is, I would not call validation. The term "evaluation" mentioned in the title is a better description. In the title, on the other hand, only GOME-2A is mentioned, while the authors are evaluating SBUV and GTO-ECV in exactly the same way. I suggest to change the title to "The use of QBO, ENSO and NAO perturbations in the evaluation of long-term total ozone satellite measurements." and to use 'evaluation' instead of 'validation' throughout the text.

**Answer to general comment 1:**

The aim of the paper can be found in the Introduction and reads as follows: "The objective of the present work is to examine the ability of the GOME-2A total ozone data to capture the variability related to dynamical proxies of global and regional importance such as the QBO, ENSO and NAO, in comparison to GB measurements, other satellite data and model calculations. The variability of total ozone from GOME-2A is compared with the variability of total ozone from the other examined data sets during these naturally-occurring fluctuations in order to evaluate the performance of GOME-2A to depict natural perturbations. The analysis is performed in the frame of the validation strategy of GOME-2A data on longer time scales within the project of EUMETSAT, AC SAF. The evaluation of GOME-2A data performed here includes the study of monthly means of total ozone, the annual cycle of total ozone, the annual cycle of total ozone, the annual cycle of the annual cycle [i.e., (max-min)/2], the relation with the QBO (zonal winds at the equator at 30 hPa), the relation with ENSO (correlation with SOI) and the relation with the NAO (correlations with the NAO index in winter (DJF mean))."

The abstract now states "Comparison of GOME-2A total ozone with ground observations shows mean differences of about  $-0.7 \pm 1.4\%$  in the tropics (0-30 deg.), about  $+0.1 \pm 2.1\%$  in mid-latitudes (30-60 deg.), and about  $+2.5 \pm 3.2\%$  and  $0.0 \pm 4.3\%$  over the northern and southern high latitudes (60-80 deg.), respectively.". Additional comparisons with ground observations are mentioned in the abstract in different lines as follows: "Differences between deseazonalised GOME-2A and GB total ozone in the tropics are within  $\pm 1\%$ .", "Differences between GOME-2A and GB measurements at the station of Samoa (American Samoa; 14.25° S, 170.6° W) are within  $\pm 1.9\%$ .", "We find very good agreement between GOME-2A and GB observations over Canada and Europe as to their NAO-related variability, with mean differences reaching the  $\pm 1\%$  levels".

While we analyse other satellite data as well, we give emphasis to GOME-2A. We prefer to keep the title as is.

We now use the term 'evaluation' instead of 'validation' throughout the text.

**General comment 2**

Throughout the paper correlations are calculated for the comparisons, which I think is very limited. I suggest that the authors provide more information on these comparisons by calculating the regression coefficients.

Answer to general comment 2:

The reviewer asks more information on the comparisons throughout the paper, which is now provided with the regression coefficients as suggested. The regression coefficients for the comparisons are presented in the new Tables 2, 3, 5, 8 (see also answer to comment 8). In addition, in the Supplement of this study we provide global maps of the regression coefficients of QBO, solar cycle, ENSO and NAO, in the Figures S1 (for QBO), S2 (for solar cycle), S3 (for ENSO) and S4 (for NAO), respectively.

**Detailed comments:**

*Comment 1: Line 23: validating => evaluating*

Answer to 1: Done

Comment 2: Line 29: Here the GTO-ECV data set is mentioned for the first time. I don't think most readers will have a clear idea what "GOME-type Total Ozone Essential Climate Variable" mean. A short description to describe this data set would be helpful at this point.

Answer to 2: We now write "... GOME-type Total Ozone Essential Climate Variable (GTO-ECV; composed of total ozone observations from GOME (Global Ozone Monitoring Experiment), SCIAMACHY (SCanning Imaging Absorption SpectroMeter for Atmospheric CHartographY), GOME-2A, and OMI (Ozone Monitoring Instrument) combined into one homogeneous time series) ..."

Comment 3: Line 51: Cause & effect are reversed in this sentence. Ozone is considered a greenhouse gas because it warms the Earth's surface not the other way around. In addition, it might be good to mention that not only tropospheric ozone but also stratospheric ozone is a greenhouse gas.

Answer to 3: The line has been revised and now reads as "In addition, ozone is a greenhouse, warming the Earth's surface. In both the stratosphere and the troposphere, ozone absorbs infrared radiation emitted from Earth's surface, trapping heat in the atmosphere. As a result, increases or decreases in stratospheric or tropospheric ozone induce a climate forcing (Hegglin et al., 2015)."

Comment 4: Line 56: "launched in 2018." => "launched end of 2018"

Answer to 4: Changed to "launched on 7 November 2018".

Comment 5: Line 73: Except for the abstract, this is the first time that the SBUV and GTO-ECV datasets are mentioned, therefore, I suggest to add references for both data sets in the text.

Answer to 5: The reference (McPeters et al., 2013) has been added here for SBUV and the references (Coldewey-Egbers et al., 2015; Garane et al., 2018) have been added for GTO-ECV.

Comment 6: Line 89-91: It might be better to refer to more recent papers about the recovery of the ozone layer, for example de Laat et al., Onset of Stratospheric Ozone Recovery in the Antarctic Ozone Hole in Assimilated Daily Total Ozone Columns, JGR, 2017, https://doi.org/10.1002/2016JD025723

Answer to 6: We have added more recent papers about the recovery of the ozone layer, as follows: Solomon et al., 2016; de Laat et al., 2017; Kuttippurath and Nair, 2017; Pazmiño et al., 2018; Stone et al., 2018; Strahan and Douglass, 2018.

The following six papers have been added in list of the references:

Solomon, S., Ivy, D. J., Kinnison, D., Mills, M. J., Neely III, R. R., and Schmidt, A.: Emergence of healing in the Antarctic ozone layer, Science, 30, doi: 10.1126/science.aae0061, 2016.

de Laat, A. T. J., van Weele, M., and van der A., R. J.: Onset of stratospheric ozone recovery in the Antarctic ozone hole in assimilated daily total ozone columns, Journal of Geophysical Research: Atmospheres, 122, 11880-11899, https://doi.org/10.1002/2016JD025723, 2017.

Kuttippurath, J. and Nair, P. J.: The signs of Antarctic ozone hole recovery, Sci. Rep., 7, https://doi.org/10.1038/s41598-017-00722-7, 2017.

Pazmiño, A., Godin-beekmann, S., Hauchecorne, A., Claud, C., Khaykin, S., Goutail, F., Wolfram, E., Salvador, J., and Quel, E.: Multiplesymptoms of total ozone recovery inside the Antarctic vortex during austral spring, Atmos. Chem. Phys, 18, 7557–7572, 2018.

Stone, K. A., Solomon, S., and Kinnison, D. E.: On the identification of ozone recovery, Geophysical Research Letters, 45, 5158-5165, https://doi.org/10.1029/2018GL077955, 2018.

Strahan, S. E. and Douglass, A. R.: Decline in Antarctic Ozone Depletion and Lower Stratospheric Chlorine Determined From Aura Microwave Limb Sounder Observations, Geophys. Res. Lett., 45, 382–390, https://doi.org/10.1002/2017GL074830, 2018.

Comment 7: Line 150-151: When mentioning the various long-term data sets of ozone, also the Multi-Sensor Reanalysis of ozone comes to mind. This data set has also been analysed for QBO, ENSO, NAO and other perturbations in Knibbe et al., ACP, 2014 and therefore is worthwhile to include here and in the discussion at the end of section 3.3.

Answer to 7: We have added the following sentence in response to the comment: "We note here that another long-term data set which has been analysed for QBO, ENSO, NAO and

other perturbations comes from the Multi-Sensor Reanalysis (Knibbe et el., 2014), but is not examined here.". Additionally, the study by Knibbe et al., ACP, 2014 is now included in the discussion at the end of section 3.3, and has been added in the list of references.

Knibbe, J. S., van der A, R. J., and de Laat, A. T. J.: Spatial regression analysis on 32 years of total column ozone data, Atmos. Chem. Phys., 14, 8461-8482, https://doi.org/10.5194/acp-14-8461-2014, 2014.

*Comment 8: Line 223: I prefer to see more than only correlation coefficients. The regression parameters could be given here and in the remainder of the analyses.*

Answer to 8: The regression parameters for the correlations shown in Figures 1 and 2 are provided in the new Table 2. The regression parameters for the comparisons with the QBO are provided in the new Table 3. The regression parameters for the comparisons with SOI are provided in the new Table 5. The regression parameters for the comparisons with NAO in winter are provided in the new Table 8.

Comment 9: Line 239-240: This sentence seems to saying that the origin of the blue zone (i.e. small amplitude) is attributed to the small amplitude in these parts. Please, give the real origin if this is known.

Answer to 9: The sentence has been corrected and now reads as follows "Interestingly, there is pattern with small amplitude of annual cycle in the southern mid-latitudes with values of about 10-15 DU, seen in Figure 4 as a blue curved line crossing the longitudes around 60 degrees south, which points to small seasonal variations of total ozone in these parts. The seasonal increase in Antarctic is delayed by 2-3 months compared to the north polar region. Only with the breakdown of the polar vortex in late spring, i.e. at a time when the poleward transport over lower latitudes has already ceased, does a strong ozone influx occur in the Antarctic. With this delay the amplitude of the seasonal variation stays much smaller poleward of 55-60° in the south than in the north (Dütsch, 1974)."

The citation (Dütsch, 1974) has been added in the list of references: Dütsch, H. U.: The ozone distribution in the atmosphere, Can. J. Chem, 52, 1491-1504, 1974.

Comment 10: Line 259-265: This analysis was already discussed in section 3.1. Only this time the monthly mean has been subtracted which does not really change the validation. I suggest to remove this or add it in section 3.1.

Answer to 10: We have removed it.

Comment 11: Line 269: A clear phase shift in Figure 5 is mentioned for higher latitudes. Actually for SBUV I see an anti-correlation with the phase for latitudes between -10 and 10, and for GOME2 I see neither phase shift or an anticorrelation. So I would not call this a clear phase shift. A discussion about the clear differences in result of SBUV (pre 2008) and GOME-2 should be added here as well. Answer to 11: For SBUV there is no anti-correlation for latitudes between -10 and 10. The regression coefficients of QBO are all positive in the tropics and negative at higher latitudes as we show in the new Table 5, and display in the Supplement Figure S1.

The part of the text discussing the correlation with the QBO has been revised and now reads as follows:

"The line with dots superimposed on the ozone anomalies in Figure 5 shows the equatorial zonal winds at 30 hPa which were used as a proxy index to study the impact of QBO on total ozone. The general features include a QBO signal in total ozone at latitudes between  $10^{\circ}$  N and  $10^{\circ}$  S which almost matches with the phase of QBO in the zonal winds. At higher northern and southern latitudes there is a phase shift in the QBO impact on total ozone. The impact of QBO is more pronounced in the tropics and it is less pronounced in the sub-tropics and mid-latitudes. Strong positive correlations with the QBO are found in the tropics (correlation between GOME-2A and QBO of about +0.77, t-test = 12.91) and weaker (usually of opposite sign) less significant correlations are found at higher latitudes (about -0.15 in the northern and about -0.45 in the southern extra tropics). Similar correlations suggest that the variability that can be attributed to the QBO is about 60% in the tropics, and about 2% and 20% in the northern and the southern extra tropics, respectively.

Table 3 summarizes the correlation and regression coefficients between total ozone and QBO at 30 hPa for the different latitude zones and the different datasets. The numbers speak for themselves: for latitudes between 10° N and 10° S correlations between total ozone from GOME-2A, GTO-ECV, SBUV, GB data and the QBO are all positive. At latitudes between 10° and 30° the correlations turn to negative, in agreement with Knibbe et al. (2014) results, who noted that moving from the tropics towards higher latitudes the regression estimates switch to negative values at approximately 10° N and 10° S. The correlations with the QBO at 30 hPa remain negative up to 60°, a consistent result among all our data sets, something also reported by Knibbe et al. (2014) with the MSR ozone data. The correlation and regression coefficients between GOME-2A and QBO are fairly similar to those found between SBUV and QBO, as well as among all data sets as seen in Table 3, despite the different periods of records."

**Comment 12: Line 291-292: The correlations are not removed but the relation between ozone and QBO has been removed. Please, reformulate.**

Answer to 12: We have reformulated as follows: "To examine the impact of ENSO on total ozone we first removed variability related to the QBO and the solar cycle, and then performed the correlation analysis with the SOI".

**Comment 13: Line 295: If you are using this equation, it would be very interesting to mention also the fitted a0 and a1 instead or in addition to the found correlations.**

Answer to 13: The fitted a0 and a1 are provided in addition to the found correlations, as follows: "The QBO-related coefficients *a*0 and *a*1 of Eq. (1) for the deseasonalized GOME-2A, GTO-ECV, TOMS/OMI/OMPS and Oslo CTM3 zonal mean data are presented in Table 3. Additional information for the regression coefficients *a*1 of QBO is provided in the

Supplement Figure S1, which shows the spatial distribution of the regression coefficients in latitude-longitude maps."

Comment 14: Section 3.3, Figure 8 and 9: the GOME2 values in the last 4 year of the Figures 8 and 9 show a much worse comparison than the other years in the time series. Is there any explanation for this? I miss this in the discussion of the results here.

Answer to 14: We have added it in Section 3.3 as follows: "Despite the small differences found, we note here that GOME-2A values in the last 4 years of Figures 8 and 9 slightly deviate from the other data sets, and correlate weaker with SOI than the other years in the time series. For instance, we estimate a drop in the correlation coefficient between GOME-2A and SOI at the station Samoa (+0.58 in the period 2007-2012 and +0.47 in the period 2007-2016), which nevertheless does not alter the statistical significance of the correlation."

*Comment 15: Line 367: A discussion of a comparison with the work of Knibbe et al., ACP, 2014 would be useful at this point.*

Answer to 15: We have added the following sentence at this point "Our results are also in agreement with Knibbe et al. (2014) who showed negative ozone effects of El Niño between  $25^{\circ}$  S and  $25^{\circ}$  N, especially over the Pacific."

Comment 16: Line 370: Here the effects of QBO are removed, but what about the ENSO perturbations? Are these also removed before continuing studying the NAO effects. The two effects have to be separated.

Answer to 16: The effect of ENSO is now removed before continuing studying the NAO effects. The new line now reads "The residuals from Eq. (3), free from seasonal, QBO, solar and ENSO related variations, were later used to study the correlation between total ozone and NAO in winter". Tables and figures 7-12 for ENSO and NAO have been revised accordingly.

Comment 17: Line 293-393: Same as previous remark.

Answer to 17: We now separate the effects using different regressions, one regression to account for the effect of QBO (Eq. 1), a second regression to account for the effect of solar cycle (Eq. 2) and a third regression to account for the effect of ENSO (Eq. 3). Variability related to ENSO is now removed with Eq. (3) before continuing studying the NAO effects. The related text, tables, and figures have been revised accordingly.

Comment 18: Line 469: This is not a real validation because a lot is still unknown about the quantification of the QBO, ENSO and NAO, therefore it is qualitative evaluation not a quantitative validation resulting in uncertainty estimates.

Answer to 18: We have corrected the text to read "to qualitatively evaluate GOME-2A" instead of "validating GOME-2A".

*Comment 19: Figure 1: It is very difficult to distinguish the GOME2-A line and the SBUV-line. The legend doesn't seem to be correct either?*

Answer to 19: The figures 1 and 2 have been redrawn using a different combination of colors.

**Answers to Comments of Reviewer 2**

We would like to thank the reviewer for the fruitful comments and suggestions which helped improving the manuscript.

**Specific comments**

Comment 1: Figures 1 and 2: Maybe you could select a different combination of colors since now it is hard the differences between the different datasets to be distinguished. Alternatively you could plot the monthly differences, instead of the actual total ozone column values.

Answer to 1: The figures 1 and 2 have been redrawn using a different combination of colors.

Comment 2: Page 6 lines 214-215: It is mentioned that the highest differences are found over the southern high latitudes, however from Figures 1 and 2 it is depicted that these are presented over the Northern high latitudes (60 - 80 N) and the highest variability (standard deviation of the mean difference) is observed over the latitude belt (60 - 80 S). In addition, these differences (especially at the high latitudes) can be affected by the fact that you have not used the same days for the construction of the monthly mean values for the different datasets.

Answer to 2: The lines have been revised as suggested, and now read as follows: "In summary, the largest differences between GOME-2A, SBUV (v8.6) and GB measurements are found over the northern high latitudes  $(60^{\circ}-80^{\circ} \text{ N})$  and the highest variability (standard deviation of the mean difference) is observed over the latitude belt  $(60^{\circ}-80^{\circ} \text{ S})$ . In addition, these differences (especially at the high latitudes) can be affected by the fact that not always the same days have been used for the construction of the monthly mean values for the different datasets."

*Comment 3: Page 7 lines 220-226: Which statistical test did you use to check the statistical significance?*

Answer to 3: We have added this sentence in the text which explains it: "The statistical significance of the correlation coefficients, R, was calculated using the *t*-test formula for R with N-2 degrees of freedom, as used in Zerefos et al. (2018)."

The formula is:

$$t = R \sqrt{\frac{N-2}{1-R^2}}$$

The citation Zerefos et al. (2018) has been added in the list of references:

Zerefos, C. S., Kapsomenakis, J., Eleftheratos, K., Tourpali, K., Petropavlovskikh, I., Hubert, D., Godin-Beekmann, S., Steinbrecht, W., Frith, S., Sofieva, V., and Hassler, B.: Representativeness of single lidar stations for zonally averaged ozone profiles, their trends and attribution to proxies, Atmos. Chem. Phys., 18, 6427-6440, https://doi.org/10.5194/acp-18-6427-2018, 2018.

Comment 4: Page 8 lines 269 – 271: I don't think that you see the amplitude of QBO effect on your total ozone column. The times series are just deseasonalized, but still contain the effect of other signals such as the 11 year solar cycle, ENSO etc and thus not all the variation can be attributed to QBO.

Answer to 4: We agree that not all the variation can be attributed to QBO, and we have revised the part of the text describing the correlation with the QBO as follows:

"The line with dots superimposed on the ozone anomalies in Figure 5 shows the equatorial zonal winds at 30 hPa which were used as a proxy index to study the impact of QBO on total ozone. The general features include a QBO signal in total ozone at latitudes between  $10^{\circ}$  N and  $10^{\circ}$  S which almost matches with the phase of QBO in the zonal winds. At higher northern and southern latitudes there is a phase shift in the QBO impact on total ozone. The impact of QBO is more pronounced in the tropics and it is less pronounced in the sub-tropics and mid-latitudes. Strong positive correlations with the QBO are found in the tropics (correlation between GOME-2A and QBO of about +0.77, t-test = 12.91) and weaker (usually of opposite sign) less significant correlations are found at higher latitudes (about -0.15 in the northern and about -0.45 in the southern extra tropics). Similar correlations suggest that the variability that can be attributed to the QBO is about 60% in the tropics, and about 2% and 20% in the northern and the southern extra tropics, respectively.

Table 3 summarizes the correlation and regression coefficients between total ozone and QBO at 30 hPa for the different latitude zones and the different datasets. The numbers speak for themselves: for latitudes between 10° N and 10° S correlations between total ozone from GOME-2A, GTO-ECV, SBUV, GB data and the QBO are all positive. At latitudes between 10° and 30° the correlations turn to negative, in agreement with Knibbe et al. (2014) results, who noted that moving from the tropics towards higher latitudes the regression estimates switch to negative values at approximately 10° N and 10° S. The correlations with the QBO at 30 hPa remain negative up to 60°, a consistent result among all our data sets, something also reported by Knibbe et al. (2014) with the MSR ozone data. The correlation and regression coefficients between GOME-2A and QBO are fairly similar to those found between SBUV and QBO, as well as among all data sets as seen in Table 3, despite the different periods of records."

*Comment 5: Figures 5 and 6: You could possible superimpose the QBO proxy on the ozone anomalies.*

Answer to 5: The QBO proxy is now superimposed on the ozone anomalies.

Comment 6: Section 3.3: You removed the effect of the annual cycle and QBO, before you correlate your ozone time series with ENSO but the effect of solar cycle could also affect your results.

Answer to 6: We now remove the effect of solar cycle and repeat our calculations. We account for the solar cycle effect in ozone, using the 10.7 cm wavelength solar radio flux

(F10.7) as a proxy, taken from the National Research Council and Natural Resources Canada at ftp://ftp.geolab.nrcan.gc.ca/data/solar\_flux/monthly\_averages/solflux\_monthly\_average.txt We used the absolute values of F10.7. The text, tables, and figures 7-12, have been revised accordingly.

**Comment 7: Page 9 lines 306-307: Which statistical test did you use for checking the statistical significance?**

Answer to 7: We used the *t*-test for *R* with *N*-2 degrees of freedom (see answer to comment 3). We have corrected the sentence as follows: "These correlations were tested as to their statistical significance in the period 2007-2016 using the *t*-test for *R* with *N*-2 degrees of freedom (as in Zerefos et al., 2018), and were found to be statistical significant."

Comment 8: Section 3.4: Here you discuss the correlations between total ozone column and the NAO during winter months, evaluating the known anti-correlation between those two factors. Maybe it would be of interest to look also the correlations during summer, following the study of Osso et al. who reported a reversal in the correlation pattern between NAO and TOC from winter to summer for southern Europe.

Ossó A, Sola Y, Bech J, Lorente J (2011) Evidence for the influence of the North Atlantic Oscillation on the total ozone column at northern low latitudes and midlatitudes during winter and summer seasons. J Geophys Res Atmos 116:D24122. doi: 10.1029/2011JD016539

Answer to 8: We have also looked at the correlations during summer, which appear in the new Figure 13 for southern Europe. The new Figure A2 of Appendix A shows the correlations in global maps. The results are discussed at the end of section 3.4 as follows:

"The anti-correlation between total ozone column and the NAO index during winter also applies to southern Europe and the Mediterranean. Following the study of Ossó et al. (2011) who reported a reversal in the correlation pattern between NAO and total ozone from winter to summer in southern Europe, we have looked at the correlations during summer as well. Figure 13 presents the comparisons for 21 ground-based stations located in the region bounded by latitudes  $30^{\circ}$ - $47^{\circ}$  N and by longitudes  $10^{\circ}$ W- $40^{\circ}$ E. Figure 13a shows results for the summer and Figure 13b shows results for winter. As evident, the anti-correlation between GB total ozone and NAO in winter (R= -0.43, slope= -0.980, t-value= -2.095, p-value= 0.0499, N = 21) reverses sign and becomes positive in the summer (R= +0.60, slope= 0.874, t-value= 3.309, p-value= 0.0037, N= 21), indicating that the NAO explains about 36% of ozone variability in the summer in this region. A similar picture is also seen from GOME-2A, GTO-ECV and SBUV data."

**Typos:**

*Page 5, line 146: 50 -> 5o* Answer: Done

*Page 5, line 149: all offsets where -> all offsets were* Answer: Done

*Page 5, line 179: we made use of the monthly -> we used the monthly* Answer: Done

*Page 6, line 181: we made use of the monthly -> we used the monthly* Answer: Done

Page 6, lines 187 – 190: "Use was made of the principal …" doesn't sound very nice maybe you could change to: "The principal component (PC)-based NAO index (DJF) provided by the … (last access: 15 June 2018) was used (or analyzed). Answer: Changed as suggested.

*Page 6, line 190: After dynamical variability add* "," Answer: Done

Page 6, line 192: The impact of tropopause variability on -> The impact of the tropopause height variations on Answer: Done

**The use of QBO, ENSO and NAO perturbations in the 1 evaluation of GOME-2/MetopA total ozone measurements 2**

Kostas Eleftheratos1,2, Christos S. Zerefos2,3,4,5, Dimitris S. Balis6, Maria-Elissavet Koukouli6, 3

John Kapsomenakis3, Diego G. Loyola7, Pieter Valks7, Melanie Coldewey-Egbers7, Christophe Lerot8, Stacey M. Frith9, Amund Søvde S. Haslerud10, Ivar S. A. Isaksen10,11, Seppo Hassinen12 4

5

1Laboratory of Climatology and Atmospheric Environment, Faculty of Geology and Geoenvironment, National and 6

- 7 Kapodistrian University of Athens, Greece
- 8 2Biomedical Research Foundation of the Academy of Athens, Athens, Greece
- 3Research Centre for Atmospheric Physics and Climatology, Academy of Athens, Athens, Greece 9
- 10 4Mariolopoulos-Kanaginis Foundation for the Environmental Sciences, Athens, Greece
- 5Navarino Environmental Observatory (N.E.O.), Messinia, Greece 11
- 6Laboratory of Atmospheric Physics, Department of Physics, Aristotle University of Thessaloniki, Greece 12
- 13 7Institut für Methodik der Fernerkundung (IMF), Deutsches Zentrum für Luft- und Raumfahrt (DLR),
- 14 Oberpfaffenhofen, Germany
- 15 8Royal Belgian Institute for Space Aeronomy (BIRA), Brussels, Belgium
- 16 9Science Systems and Applications, Inc., Lanham, MD, USA
- 10Cicero Center for International Climate Research, Oslo, Norway 17
- 11Department of Geosciences, University of Oslo, Oslo, Norway 18
- 12Finnish Meteorological Institute, Helsinki, Finland 19
- 20 Correspondence to: Kostas Eleftheratos (kelef@geol.uoa.gr)

[revised manuscript text omitted]

310 These features are also evident in Figure 6 which compares GOME-2A (and GTO-ECV) satellite total ozone with 311 GB observations with respect to the QBO. Mean differences and standard deviations between GOME-2A and GB 312 and between GTO-ECV and GB deseasonalised total ozone data do not exceed one percent (Table 2). Again, 313 correlation coefficients between deseasonalised GOME-2A and deseasonalised GB data are highly significant in all 314 latitude zones (30°-60° N: +0.91 (slope=0.818, error=0.035, t-value=23.466, N=119); 10°-30° N: +0.91 315 (slope=0.786, error=0.033, t-value=23.529, N=119; 10° N-10° S: +0.94 (slope=0.973, error=0.034, t-value=28.449, 316 N=109; 10°-30° S: +0.87 (slope=0.864, error=0.044, t-value=19.659, N=119; 30°-60° 
[revised manuscript text omitted]

- 773

---

## Author Response (AR2)

Dear Editor,

We have added in the captions of Figs. S3 and S4 (Supplement) the explanation for the markers/crosses in the plots. Thank you very much.

Kind regards
Kostas Eleftheratos